**STEM CELLS AND REGENERATION**

# Myc and Tor drive growth and cell competition in the regeneration blastema of *Drosophila* wing imaginal discs

Felicity Ting-Yu Hsu and Rachel K. Smith-Bolton*

## ABSTRACT

During the regeneration of injured or lost tissues, the regeneration blastema serves as a hub for robust growth. *Drosophila* imaginal discs are a genetically tractable and simple model system for the study of regeneration and organization of this regrowth. Key signals that contribute to regenerative growth in these discs, such as reactive oxygen species, Wnt/Wg, JNK, p38, JAK/STAT and the Hippo pathway, have been identified. However, a detailed exploration of the spatial organization of regrowth, the factors that directly drive this growth, and the consequences of activating drivers of regeneration has not been undertaken. Here, we find that regenerative growth in imaginal discs is controlled by the transcription factor Myc and by Tor signaling, which drive proliferation and translation in the regeneration blastema. The spatial organization of growth in the blastema is arranged into concentric growth zones defined by Myc expression, elevated Tor activity and elevated translation. In addition, the increased Myc expression in the innermost zone induced Xrp1-independent cell competition-like death in the adjacent zones, revealing a delicate balance between driving growth and inducing death in the regenerating tissue.

KEY WORDS: Regeneration, *Drosophila*, Myc, Tor, Cell competition

## INTRODUCTION

Regeneration is a phenomenon through which some animals restore tissues or organs after damage. This regrowth often occurs through proliferation of a group of undifferentiated and robustly growing cells at the damage site, called a blastema (reviewed by Londono et al., 2018). To study tissue regeneration, we use *Drosophila* wing imaginal discs as a model. The ability of these appendage primordia to regenerate during the larval stage was discovered when the imaginal discs were cut and incubated in the abdomens of female flies (Hadorn and Buck, 1962). Two decades later, further studies revealed blastema-like cells robustly proliferating at the wound site in leg and wing imaginal discs (Abbott et al., 1981; O'Brochta and Bryant, 1987). Subsequent searches for signaling pathways important for regenerative growth identified Jun N-terminal Kinase (JNK) signaling as crucial for disc regeneration after fragmentation (Bosch et al., 2005), and lineage tracing demonstrated that cells experiencing

Cell and Developmental Biology, University of Illinois at Urbana-Champaign, Urbana, IL 61801, USA.

*Author for correspondence (rsbolton@illinois.edu)

 R.K.S.-B., 0000-0003-2196-8275

JNK signaling were the origin of the regeneration blastema (Bosch et al., 2008). More recently, genetic ablation systems have replaced fragmentation in regeneration studies; such systems generate localized massive cell death through overexpression of pro-apoptotic genes (Fox et al., 2020). In these ablation systems, blastema cells were also found around the wound sites during regeneration (Smith-Bolton et al., 2009). To better define the blastema, two research groups (Floc'hlay et al., 2023; Worley et al., 2022) used single-cell RNA-sequencing (scRNA-seq) to characterize blastema cells by gene expression. This approach identified two clusters of cells with expression of signaling molecules such as Wg and Upd3 and transcription factors such as Ets21C that define the blastema (Floc'hlay et al., 2023; Worley et al., 2022). However, gene expression alone may not capture all changes in cell behavior and may not identify all cells contributing to the regenerating tissue.

During normal development, low ecdysone levels maintain larval development and growth of the imaginal discs, while a burst of high ecdysone production triggers developmental transitions, such as molting or pupariation (Thummel, 2001). The ability of imaginal discs to form a blastema is reduced during the late third instar due to an increase in the hormone ecdysone, which silences damage-responsive signals (Halme et al., 2010; Harris et al., 2016; Jaszczak and Halme, 2016). Damage during the early- and mid-third larval instar induces secretion of the relaxin-like peptide *Drosophila* insulin-like peptide 8 (dILP8), which reduces systemic ecdysone and delays the surge to prolong larval development. This reduced ecdysone also reduces imaginal disc growth outside of the damaged area (Colombani et al., 2012; Garelli et al., 2012; Halme et al., 2010; Jaszczak and Halme, 2016; Jaszczak et al., 2015, 2016; Karanja et al., 2022; Katsuyama et al., 2015). Therefore, the regeneration blastema must over-ride the global reduction of growth due to low ecdysone levels.

Here, we demonstrate that the transcription factor Myc and the Target of Rapamycin (Tor) signaling pathway drive blastema growth. Both Myc transcription and Tor signaling were upregulated during regeneration. Reducing their expression or kinase activity reduced proliferation and translation in the blastema. Interestingly, the area with activated Tor signaling extended beyond the area with high Myc expression, establishing concentric zones of blastema growth.

Cell competition occurs when neighboring cells have different fitness levels, and 'loser cells' are eliminated by cell death, while 'winner cells' replace the loser cells. Differences in Myc expression between adjacent cell populations lead to cell competition or super-competition during normal development (de la Cova et al., 2004, 2014; Meyer et al., 2014; Morata, 2021). More specifically, cell competition occurs when wild-type cells are winners and cells with reduced Myc expression are losers, and super-competition occurs when cells with elevated Myc expression are winners and wild-type cells are losers (reviewed by Morata, 2021). Death of the loser cells can involve autophagy (Nagata et al., 2019) and expression of the transcription factor Xrp1 (Baillon et al., 2018; Blanco et al., 2020; Kiparaki et al., 2022; Lee et al., 2018; Ochi et al., 2021).

Proteotoxic stress, as assessed by increased levels of phosphorylated eIF2α, has been reported in loser cells during cell competition (Baumgartner et al., 2021), and phosphorylated eIF2α is sufficient to induce loser cell fate (Kiparaki et al., 2022). However, increased phosphorylation of eIF2α also occurs in winner cells during super-competition, although these cells are protected from this proteotoxic stress and do not undergo apoptosis (Paul et al., 2024). Thus, eIF2α phosphorylation can be used to distinguish between cell competition and super-competition. In addition, several mechanisms can play a role in the comparison of cell fitness during various forms of cell competition (Morata, 2021; Nagata and Igaki, 2024), including Spätzle/Toll signaling (Alpar et al., 2018; Germani et al., 2018; Meyer et al., 2014) and Flower/Azot signaling (Merino et al., 2015; Rhiner et al., 2010). These mechanisms that govern cell competition in wing imaginal discs have predominantly been studied using induced clones of genetically distinct cell populations (reviewed by Khandekar and Ellis, 2024). Whether the Myc expression differences that arise during regeneration also lead to competitive behaviors and whether the underlying mechanism aligns with any of the established models of cell competition were unclear.

Here, we show that cell death is increased in the cells adjacent to the high Myc-expressing cells in the blastema. Furthermore, this cell death was suppressed when Myc expression was eliminated, indicating the presence of cell competition. The cells adjacent to the high-Myc cells showed an increase in autophagy and eIF2α phosphorylation, similar to loser cells in cell competition. However, we did not find elevated Xrp1 expression, and heterozygosity for Xrp1 did not eliminate Myc-induced cell death in regenerating wing discs. In addition, the known fitness comparison mechanisms were not active. Therefore, the heterogeneity and complex signaling in the regeneration blastema likely leads to a type of cell competition that is not identical to the competition induced when two sets of cells differ by only one factor. The finding that wing disc regeneration involves not only robust regrowth but also increased cell death suggests that there is a dynamic balance between growth and death, providing new insights into the microenvironment of regenerating tissue.

## RESULTS
### Myc and Tor drive blastema growth
To study regeneration in the *Drosophila* wing imaginal disc, we used a genetic ablation system in which tissue damage can be induced with spatial and temporal precision (Smith-Bolton et al., 2009). This ablation system uses the UAS/Gal4 transcriptional system, where a Gal4 in the *rotund* (*rn*) locus directs the expression of the pro-apoptotic gene *reaper* (*rpr*) in the wing imaginal disc pouch (Fig. 1A). In addition, the Gal4 activity is constrained by Gal80[ts] at 18°C but not at 30°C. We raised the larvae in an 18°C incubator for 7 days after egg-laying, when the larvae reach the early third instar stage, then transferred the larvae to a 30°C water bath for 24 h to activate apoptosis. After the 24-h temperature shift, the larvae were returned to 18°C. This time point was considered recovery time (R0) (Fig. 1A).

To characterize growth in a regenerating wing disc, we examined proliferation in undamaged and regenerating discs 24 h after the end of tissue damage, or R24. We used EdU incorporation to mark cells in S phase and anti-phospho-histone H3 (PH3) immunostaining to mark cells in mitosis. We compared the undamaged wing pouch, the outer edge of which is marked by an inner ring of Wg expression (Fig. S1A,A′) (Swarup and Verheyen, 2012), and the regeneration blastema, which is marked by Wg expression throughout (Fig. S1B,B′) (Smith-Bolton et al., 2009; Worley et al., 2022). In undamaged wing discs, cells in S phase were distributed evenly throughout the wing disc

(Fig. 1B). However, after tissue damage, the blastema exhibited increased EdU incorporation (Fig. 1C,D), suggesting more cells were in S phase. Similarly, mitotic cells appeared uniform throughout an undamaged wing disc (Fig. 1E), but the regeneration blastema had an increased number of mitotic cells (Fig. 1E-G).

Next, we investigated whether regeneration also involves increased growth driven by protein translation. To examine protein translation, we used the O-propargyl-puromycin (OPP) assay, which marks newly synthesized protein by incorporating a puromycin analog detected by click chemistry. In undamaged discs, translation seemed evenly distributed throughout (Fig. 1H). However, after tissue damage, the blastema had markedly increased translation (Fig. 1I,J).

Given that proliferation and translation were elevated in the regeneration blastema, we were interested in what drives this robust growth. To answer this question, we examined Ecdysone Receptor (EcR) activity, glycolysis, Myc expression and Tor signaling. We first examined whether blastema cells escape reduced growth caused by reduced ecdysone by upregulating activity of the EcR (Fig. S1). Indeed, our transcriptional profile of the regeneration blastema showed slight upregulation of *EcR*, a log2 increase of 1.16 (*P*=0.00005) (Khan et al., 2017). In addition, EcR activity is increased during later regeneration (R32) and is important for regeneration (Terry et al., 2024). To assess EcR activity at R24, we used the *7xEcRE-GFP* reporter, which has seven EcR response elements (EcREs) upstream of GFP (Hackney et al., 2007; Terry et al., 2024). The EcRE-GFP reporter was active throughout an undamaged wing pouch (Fig. S1C) and was not elevated in the blastema at R24 (Fig. S1D), indicating that enhanced EcR activity does not drive growth at R24.

Glycolysis is increased during early zebrafish fin regeneration due to the upregulation of lactate dehydrogenase (Ldh) (Brandão et al., 2022). Therefore, we investigated whether increased glycolysis and lactate dehydrogenase might be driving the growth in the *Drosophila* regeneration blastema. However, a reporter line for *Lactate dehydrogenase* (*Ldh*) expression, *Ldh-GFP[Genomic]* (Bawa et al., 2020; Rai et al., 2024), did not show increased expression in the blastema (Fig. S1E,F). Thus, upregulated Ldh does not drive the robust growth in the regeneration blastema.

Myc is a transcription factor that regulates cell growth by promoting proliferation and translation (Bellosta and Gallant, 2010; Grewal et al., 2005; Johnston et al., 1999) and is important for regeneration in multiple animals, including *Hydra*, axolotl, *Xenopus* and zebrafish (reviewed by Ascanelli et al., 2024). In *Drosophila* imaginal discs, Myc is highly expressed in the blastema, overexpression of Myc leads to better regeneration, and reduction of Myc causes poor regeneration (Abidi et al., 2023; Harris et al., 2020; Serras and Bellosta, 2024; Smith-Bolton et al., 2009). To determine whether Myc is upregulated through transcription, we used a *Myc-lacZ* reporter line. In undamaged discs, *Myc-lacZ* expression was observed throughout the pouch, with a slight reduction at the dorsal-ventral boundary (Fig. 1K). At R24, *Myc-lacZ* was upregulated in the regeneration blastema (Fig. 1L,M). We previously found that Myc expression was difficult to reduce during regeneration, either through RNAi knockdown or use of a heterozygous mutation (Abidi et al., 2023), suggesting that its expression is tightly regulated.

Tor regulates translation by phosphorylating ribosomal p70 S6 kinase I (S6K), which in turn phosphorylates ribosomal protein S6 (RpS6) (reviewed by Frappaolo and Giansanti, 2023). We examined Tor pathway activity through anti-phospho-RpS6 (p-S6) immunostaining (Romero-Pozuelo et al., 2017). In an undamaged wing disc, elevated p-S6 occurred in a patchy pattern (Fig. 1N), consistent with previous studies (Romero-Pozuelo et al., 2017). At

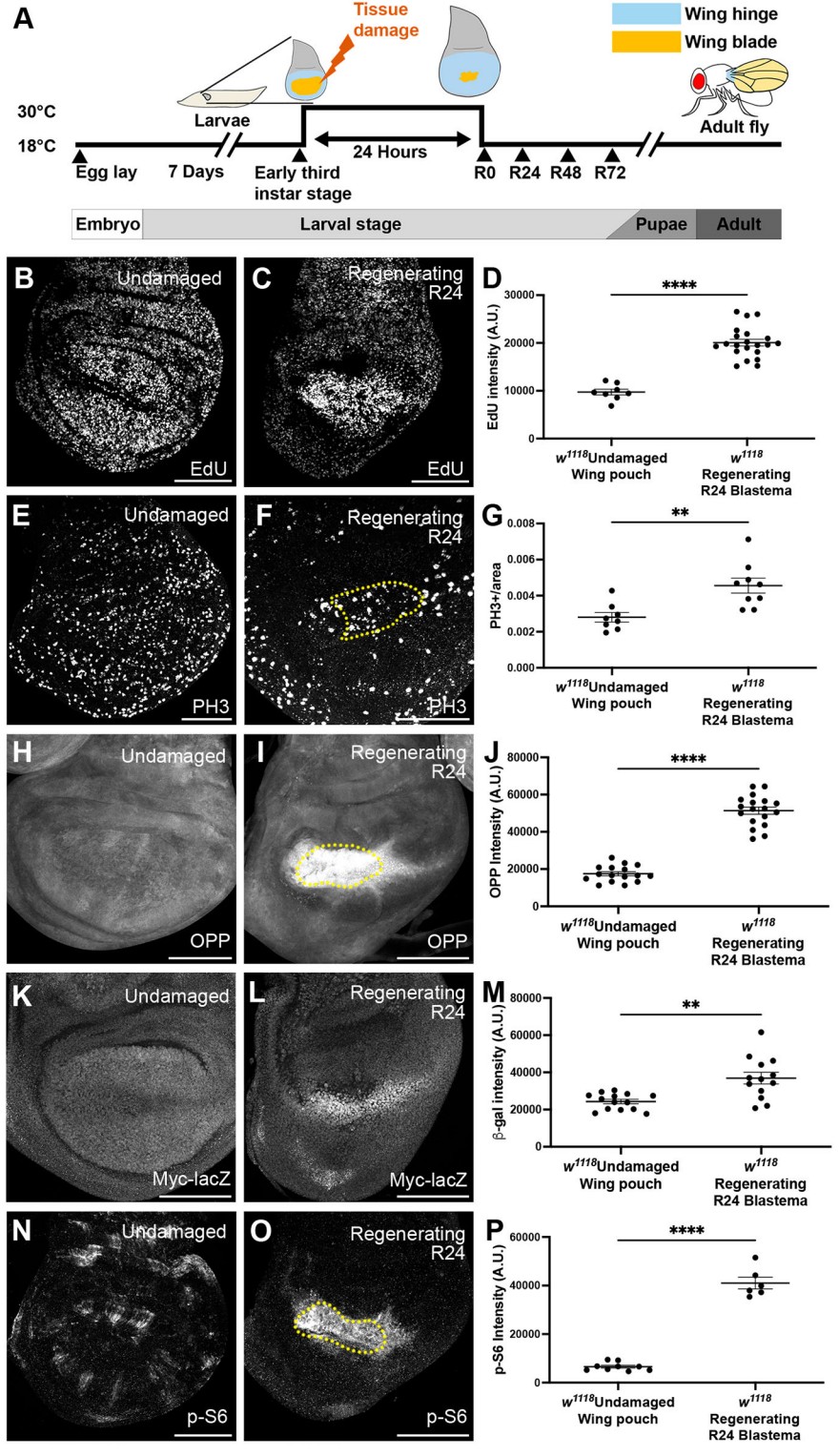

**Fig. 1. Myc expression and Tor signaling in regenerating wing discs.** (A) Schematic of the genetic ablation system. (B,C) EdU incorporation marking S-phase cells in an undamaged disc (B) and an R24 disc (C). (D) Quantification of average EdU staining intensity in the wing pouch identified by morphological features. Undamaged, $n=8$; R24, $n=21$. ****$P<0.0001$. (E,F) PH3 staining marking mitotic cells in an undamaged disc (E) and an R24 disc (F). (G) Quantification of mitotic cells per area in the wing pouch identified by inner ring of Wg staining (yellow dotted lines in F). Undamaged, $n=8$; R24, $n=9$. **$P<0.01$. (H,I) OPP assay in an undamaged disc (H) and an R24 disc (I). (J) Quantification of OPP staining intensity in the wing pouch of an undamaged disc or blastema of an R24 disc, identified by the inner ring of Wg staining or Wg staining, respectively. Undamaged, $n=16$; R24, $n=18$. ****$P<0.0001$. (K,L) *Myc-lacZ* expression detected by β-galactosidase staining in an undamaged disc (K) and an R24 disc (L). (M) Quantification of average β-gal staining intensity in the wing pouch or blastema identified by morphological features. Undamaged, $n=14$; R24, $n=13$. **$P<0.01$. (N,O) p-S6 staining in an undamaged disc (N) and an R24 disc (O). (P) Quantification of average p-S6 staining intensity in the wing pouch or blastema identified by Wg staining (yellow dotted lines in O). Undamaged, $n=9$; R24 $n=6$. ****$P<0.0001$. Statistical test used was Welch's *t*-test. Error bars are s.e.m. A.U., arbitrary units. Scale bars: 100 μm.

R24, p-S6 was enhanced in the regenerating blastema (Fig. 1O,P), suggesting that the Tor pathway might also be an important driver of growth during wing disc regeneration.

## Myc promotes regenerative growth in a damaged wing disc

To determine the extent to which Myc contributes to the growth of the blastema, we examined proliferation and protein synthesis in regenerating wing discs with reduced Myc expression. We have previously shown that animals hemizygous for the hypomorphic allele $Myc^{P0}$ exhibited substantially reduced Myc throughout the wing disc and eliminated the elevated Myc in the blastema of regenerating wing discs, leading to poor regeneration (Abidi et al., 2023) (Fig. S2A,B). By comparing 5-ethynyl-2'-deoxyuridine (EdU) incorporation in control and $Myc^{P0}$ regeneration blastemas (Fig. 2A,B), we found a reduction in EdU when Myc levels were reduced (Fig. 2C). Similarly, measurement of mitoses by anti-PH3 staining showed fewer mitotic cells in the $Myc^{P0}$ blastema (Fig. 2D-F, Fig. S2C,D). We used the OPP assay to measure protein

translation and found that $Myc^{P0}$ regenerating discs had reduced translation (Fig. 2G-I). Thus, Myc promotes growth in the regeneration blastema by promoting translation as well as proliferation.

As Myc regulates growth and translation at least partially through ribosomal biogenesis (Grewal et al., 2005), we examined ribosomal biogenesis by measuring the size of the nucleoli using an anti-Fibrillarin antibody. We measured ten nucleoli in a 200-pixel$^2$ area in the hinge and pouch or blastema of each disc. In undamaged discs, the pouch nucleoli and the hinge nucleoli were similar in size (Fig. 2J-L,V). In regenerating wing discs, the blastema cells had larger nucleoli than the hinge cells (Fig. 2M-O,V). Importantly, the

nucleoli in the blastema cells were also larger than those in undamaged discs (Fig. 2K,N,V).

To determine whether the nucleoli in the blastema were larger due to increased Myc expression, we measured nucleoli in undamaged and damaged $Myc^{P0}$ discs. In $Myc^{P0}$ undamaged discs, the nucleoli were the same size in the wing pouch and in the hinge (Fig. 2P-R,V) and smaller than those in control undamaged discs (Fig. 2V). This result was expected because Myc expression was reduced throughout the disc. In $Myc^{P0}$ regenerating discs, nucleoli in the $Myc^{P0}$ blastema were still larger than the nucleoli in the $Myc^{P0}$ hinge (Fig. 2S-U,V) but were smaller than those in the control blastema

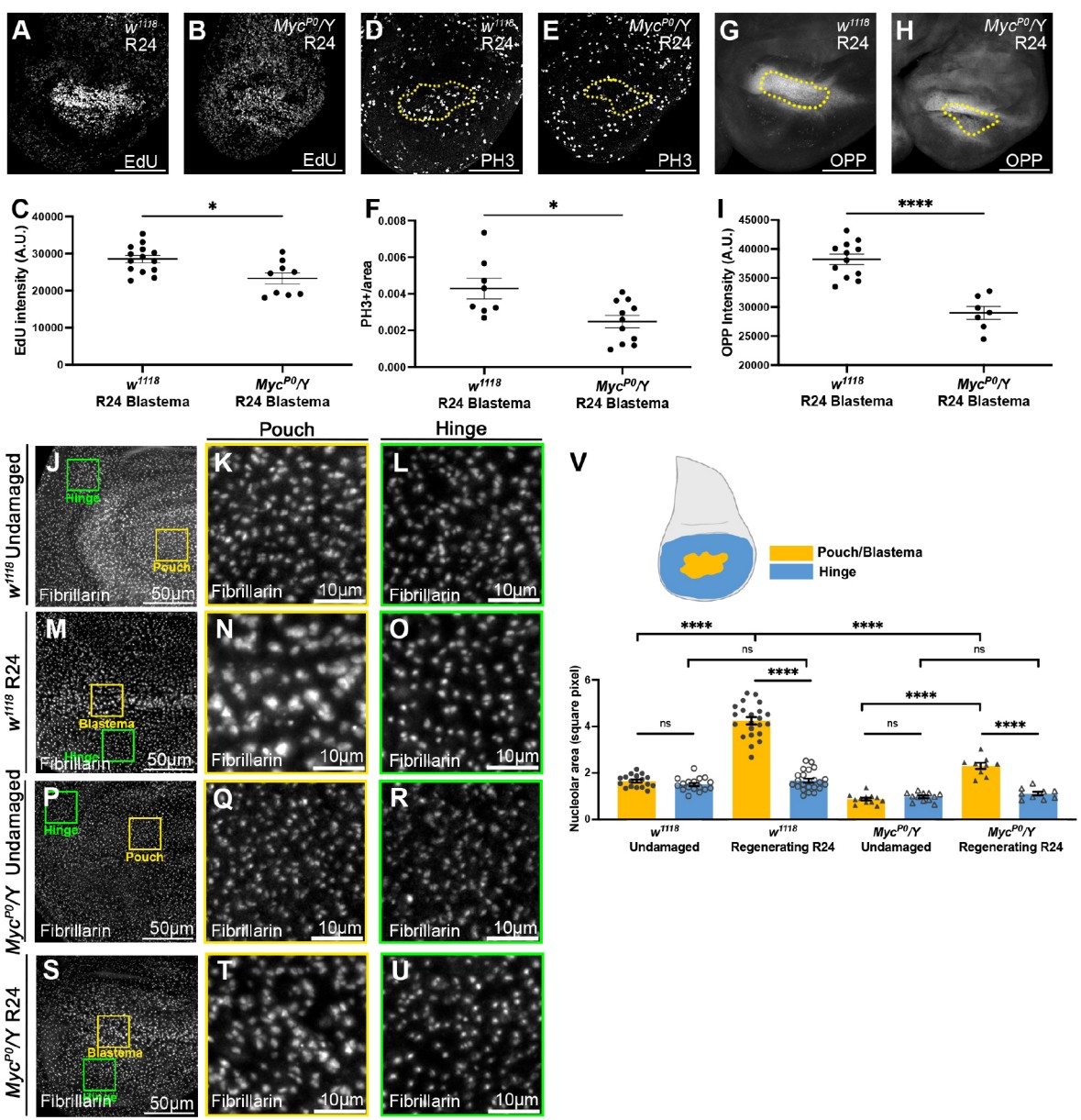

**Fig. 2. Myc promotes growth in the regeneration blastema.** (A,B) EdU incorporation in a $w^{1118}$ R24 disc (A) and a $Myc^{P0}/Y$ R24 disc (B). (C) Quantification of average EdU intensity in the wing pouch identified by morphological features. $w^{1118}$, $n$=14; $Myc^{P0}/Y$, $n$=9. *$P$<0.05. (D,E) PH3 staining in a $w^{1118}$ R24 disc (D) and a $Myc^{P0}/Y$ R24 disc (E). (F) Quantification of mitotic cells per area in the blastema (Wg; yellow dotted lines in D,E). $w^{1118}$, $n$=8; $Myc^{P0}/Y$, $n$=11. *$P$<0.05. (G,H) OPP assay in a $w^{1118}$ R24 disc (G) and a $Myc^{P0}/Y$ R24 disc (H). (I) Quantification of average OPP signal in the Wg-positive blastema (yellow dotted lines in G,H). $w^{1118}$, $n$=12; $Myc^{P0}/Y$, $n$=7. ****$P$<0.0001. (J-U) Fibrillarin staining marking nucleoli in a $w^{1118}$ undamaged disc (J-L), a $w^{1118}$ R24 disc (M-O), a $Myc^{P0}/Y$ undamaged disc (P-R) and a $Myc^{P0}/Y$ R24 disc (S-U). (K,N,Q,T) 200-pixel square from pouch or Wg-positive blastema (yellow boxes). (L,O,R,U) 200-pixel square from hinge (green boxes). (V) Quantification of nucleolus size. $w^{1118}$, undamaged $n$=17; $w^{1118}$ R24, $n$=22; $Myc^{P0}/Y$ undamaged, $n$=12; $Myc^{P0}/Y$ R24, $n$=9. ****$P$<0.0001. ns, not significant ($P$>0.05). Statistical test used was Welch's $t$-test. Error bars are s.e.m. A.U., arbitrary units. Scale bars: 100 μm.

(Fig. 2N,T,V). Therefore, Myc contributes to the increase in nucleolus size in the regeneration blastema but does not account for all of the increase in ribosomal production and nucleolus size.

## Tor promotes growth in the regeneration blastema

To determine the extent to which Tor signaling contributes to regeneration, we examined the effect of a reduction in Tor activity using animals heterozygous for $Tor^{2L1}$, a hypomorphic allele that has impaired kinase activity due to a single amino acid change in the kinase domain (Oldham et al., 2000). To confirm that Tor activity was reduced in $Tor^{2L1}/+$ regenerating discs, we assessed anti-p-S6 immunostaining at R24 (Fig. 3A,B), using Wg as a blastema marker (Fig. S3A,B). As expected, p-S6 signal was reduced in $Tor^{2L1}/+$ regenerating discs (Fig. 3A-C, Fig. S3A,B). In addition, we examined the adult wings that developed from regenerated discs of $Tor^{2L1}/+$ flies as a proxy for how well regeneration occurred in the imaginal disc. The $Tor^{2L1}/+$ flies had smaller wings after disc regeneration, indicating that the imaginal discs had regenerated poorly (Fig. S3C).

We assessed proliferation in $Tor^{2L1}/+$ regenerating discs using EdU incorporation and anti-PH3 staining (Fig. 3D-I, Fig. S3D,E). EdU intensity was reduced when Tor activity was reduced (Fig. 3D-F). Similarly, the number of mitotic cells per area was reduced when Tor activity was reduced (Fig. 3G-I, Fig. S3D,E).

We examined protein translation in $Tor^{2L1}/+$ regenerating discs using the OPP assay and found reduced protein translation in the blastema when Tor signaling was reduced (Fig. 3J-L, Fig. S3F,G). Thus, we concluded that Tor signaling promotes regeneration blastema growth by regulating both proliferation and translation. Interestingly, we noticed that the area marked by Wg expression was smaller than the area with upregulated translation (Fig. 3J,K), suggesting that regenerative growth may be elevated outside of the zone marked by expression of canonical blastema genes.

Next, we examined whether Tor enhances growth in part by enhancing ribosomal biogenesis. However, the nucleoli in the blastemas of $Tor^{2L1}/+$ discs were not smaller than those in the controls (Fig. S3H-N). We wondered whether the heterozygous $Tor^{2L1}$ allele was not strong enough for us to detect a requirement for Tor in ribosomal biogenesis. Therefore, we fed the larvae rapamycin to reduce Tor signaling (Fig. 3M-S). While the blastema cells had larger nucleoli than cells in the hinge in both control and rapamycin-fed animals (Fig. 3M,P), the rapamycin-fed animals had smaller nucleoli in the blastema compared to those in larvae fed control food (Fig. 3N,Q,S). Thus, Tor signaling is important for ribosome biogenesis during regeneration.

## Crosstalk between Tor signaling and Myc in the regeneration blastema

Several studies have shown interactions between Myc and Tor in growth regulation in *Drosophila* (Kuo et al., 2015; Parisi et al., 2011; Teleman et al., 2008). Thus, we investigated whether there is also crosstalk between Myc and Tor in the regeneration blastema. First, we examined whether regenerating discs with reduced Myc have reduced Tor activity by immunostaining for p-S6 in $Myc^{P0}/Y$ regenerating discs and found reduced p-S6 intensity (Fig. 3T-V). Thus, Myc is important for increased Tor signaling in the regeneration blastema.

Next, we examined whether reduced Tor activity would reduce Myc expression in the regeneration blastema. We used rapamycin feeding to inhibit Tor activity and the *Myc-lacZ* reporter to examine Myc transcription. We saw reduced transcription of the Myc reporter in the animals fed with rapamycin, compared to the animals fed with control food (Fig. 3W-Y), suggesting that Tor signaling regulates Myc expression. Together, these results establish the

existence of crosstalk between Myc and Tor signaling in the regeneration blastema.

## Mapping the domain of high Myc expression in the regeneration blastema

Two sets of scRNA-seq data from regenerating *Drosophila* wing imaginal discs identified cells belonging to the regeneration blastema, termed blastema 1 and blastema 2 cells (Worley et al., 2022), or wound alpha cells (Floc'hlay et al., 2023). The ablation assays used in these two studies were slightly different; we used the pro-apoptotic gene *reaper* to induce cell death whereas these studies used the TNFα *eiger*. In addition, Floc'hlay et al. activated ablation for 40 h instead of the 24 h that we and Worley et al. used. Therefore, to confirm the expression of the blastema markers in our ablation system, we used Upd3, a JAK/STAT ligand, as a blastema 1 marker and Wg as a blastema 1 and 2 marker (Worley et al., 2022). To visualize *upd3* expression, we used the reporter line *upd3-lacZ* (Bunker et al., 2015). In regenerating wing discs, we confirmed two zones in the regeneration blastema, in which the Wg-expressing area was larger than the *upd3-lacZ*-expressing area (Fig. 4A-B″), confirming the existence of blastema 1 and 2 cells in our discs. To understand where Myc is upregulated, we used Wg as a blastema marker and co-immunostained for Myc. Although Wg did not colocalize with Myc in an undamaged disc (Fig. 4C), in R24 discs high Myc expression coincided with high Wg expression, as previously reported (Smith-Bolton et al., 2009) (Fig. 4D-D″).

We also determined the extent to which the blastema as defined by Worley et al. coincided with Nubbin (Nub) expression, which is normally found in the wing pouch and overlaps with the inner ring of Wg at the boundary between the pouch and the hinge (Ng et al., 1995; Terriente et al., 2008). In an undamaged wing disc, the Nub-expressing area was slightly larger than the moderate Myc-expressing area (Fig. 4E). During regeneration, the Nub-expressing area was also larger than the high Myc-expressing area (Fig. 4F-F″). Therefore, the blastema as defined by Wg and Myc expression is smaller than the Nub-expressing area. To determine which cells in the regenerating disc were hinge cells and if there is any overlap between the hinge and the regeneration blastema, we used a Zfh2 antibody (Tran et al., 2010) (Fig. 4G-H″) and the JAK/STAT signaling reporter *10XSTAT92E-GFP* (Bach et al., 2007) (Fig. 4I-J′) as hinge markers. In both undamaged (Fig. 4G,I) and damaged (Fig. 4H-H″,J-J′) discs, the hinge markers were expressed directly adjacent to, but not overlapping with, the zone with elevated Myc expression.

To determine whether Nub and Zfh2 expression overlapped in regenerating discs, we used the *nub-MiMIC-GFP* ($nub^{MI05126}$) line, in which a MiMIC transposable element inserted into the *nub* locus expresses GFP in cells that are also marked with anti-Nub immunostaining (Khan et al., 2017; Venken et al., 2011). In undamaged wing discs, Nub-GFP expression overlapped with Zfh2-positive cells at the inner ring of Wg expression (Fig. 4K,K′), consistent with previous studies (Terriente et al., 2008). In regenerating wing discs, we also found Nub-GFP expression overlapping with Zfh2 expression (Fig. 4L,L′, Fig. S4). Furthermore, Wg, Nub and Zfh2 expression together showed that the blastema was smaller than the Nub-expressing area but directly adjacent to the cells expressing both Nub and the hinge markers (Fig. 4L,L″). Interestingly, hinge cells adjacent to the blastema are flexible in their identity and can become pouch cells during re-growth (Herrera et al., 2013; Ledru et al., 2022; Worley et al., 2018), which may be occurring in the cells expressing both Nub and Zfh2. Thus, the high Myc-expressing zone colocalized with the blastema as defined by single-cell sequencing and did not overlap with hinge cells.

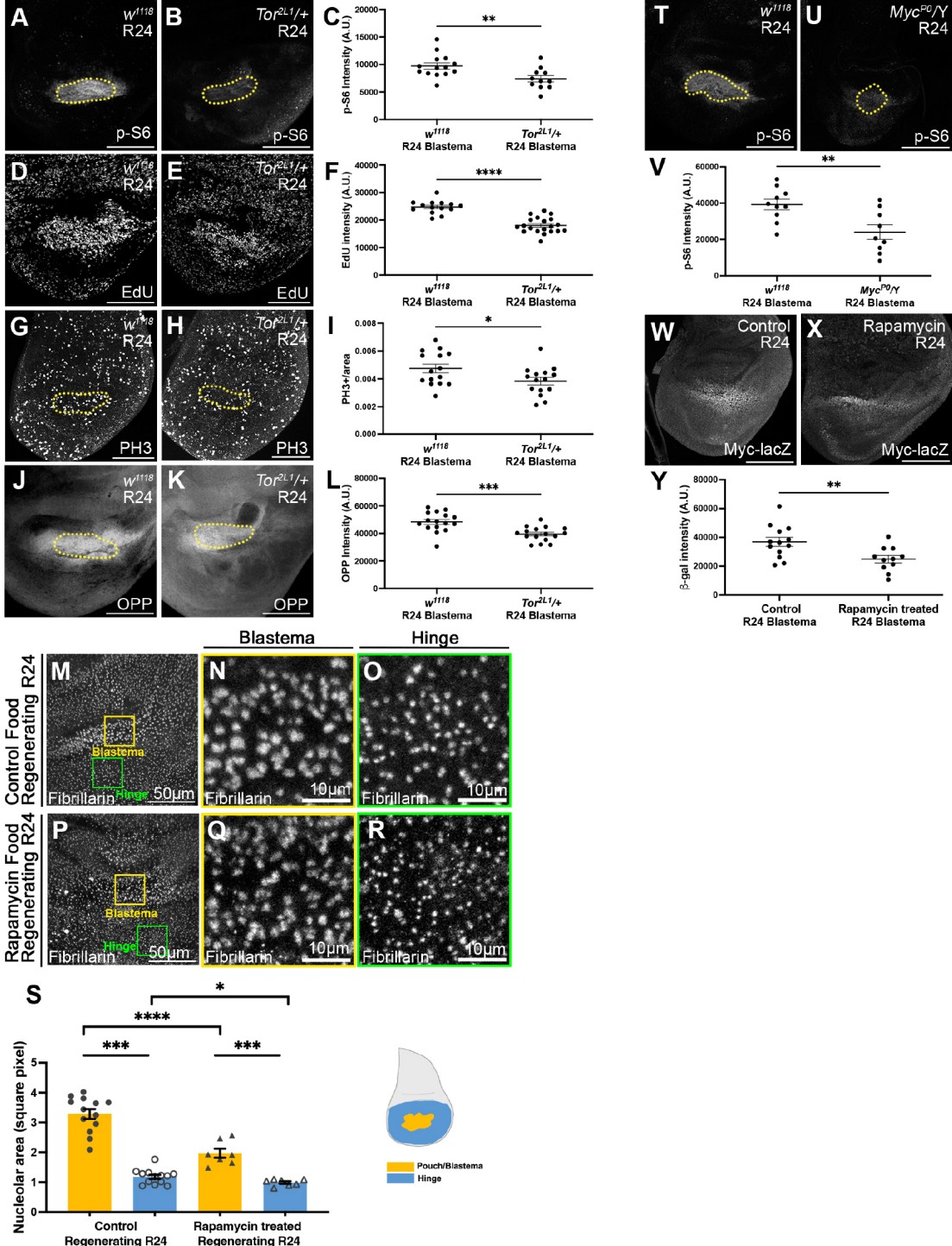

**Fig. 3. Tor promotes growth in the regeneration blastema.** (A,B) p-S6 immunostaining in a $w^{1118}$ R24 disc (A) and a $Tor^{2L1}$/+ R24 disc (B). (C) Quantification of average p-S6 staining intensity in the Wg-positive blastema (yellow dotted lines in A,B). $w^{1118}$, $n=14$; $Tor^{2L1}$/+, $n=11$. **$P<0.01$, Welch's $t$-test. (D,E) EdU incorporation in a $w^{1118}$ R24 disc (D) and a $Tor^{2L1}$/+ R24 disc (E). (F) Quantification of average EdU staining intensity in the blastema identified by morphological features. $w^{1118}$, $n=14$; $Tor^{2L1}$/+, $n=20$. ****$P<0.0001$. (G,H) PH3 staining in a $w^{1118}$ R24 disc (G) and a $Tor^{2L1}$/+ R24 disc (H). (I) Quantification of mitotic cells per area in the Wg-positive blastema (yellow dotted lines in G,H). $w^{1118}$, $n=15$; $Tor^{2L1}$/+, $n=15$. *$P<0.05$. (J,K) OPP assay in a $w^{1118}$ R24 disc (J) and a $Tor^{2L1}$/+ R24 disc (K). (L) Quantification of average OPP staining in the Wg-positive blastema (yellow dotted lines in J,K). $w^{1118}$, $n=16$; $Tor^{2L1}$/+, $n=16$. ***$P<0.001$. (M-R) Fibrillarin staining in an R24 disc after feeding with control food (M-O) or food containing rapamycin (P-R). (N,Q) 200-pixel squares from the Wg-positive blastema (yellow boxes). (O,R) 200-pixel square from the hinge (green boxes). (S) Quantification of nucleolus size in hinge and blastema. control, $n=13$; rapamycin, $n=7$. *$P<0.05$, ***$P<0.001$, ****$P<0.0001$. (T,U) p-S6 staining in a $w^{1118}$ (T) and a $Myc^{P0}$/Y (U) disc. (V) Quantification of average p-S6 staining intensity in Zone A identified by Wg staining (yellow dotted lines in T,U). $w^{1118}$, $n=10$; $Myc^{P0}$/Y, $n=9$. **$P<0.01$. (W,X) $Myc$-$lacZ$ expression marked by β-galactosidase staining in a control R24 disc (W) and a rapamycin-treated R24 disc (X). (Y) Quantification of average β-galactosidase staining intensity in the blastema. control, $n=13$; rapamycin, $n=11$. **$P<0.01$. Statistical test used was Welch's $t$-test. Error bars are s.e.m. A.U., arbitrary units. Scale bars: 100 μm (unless otherwise marked).

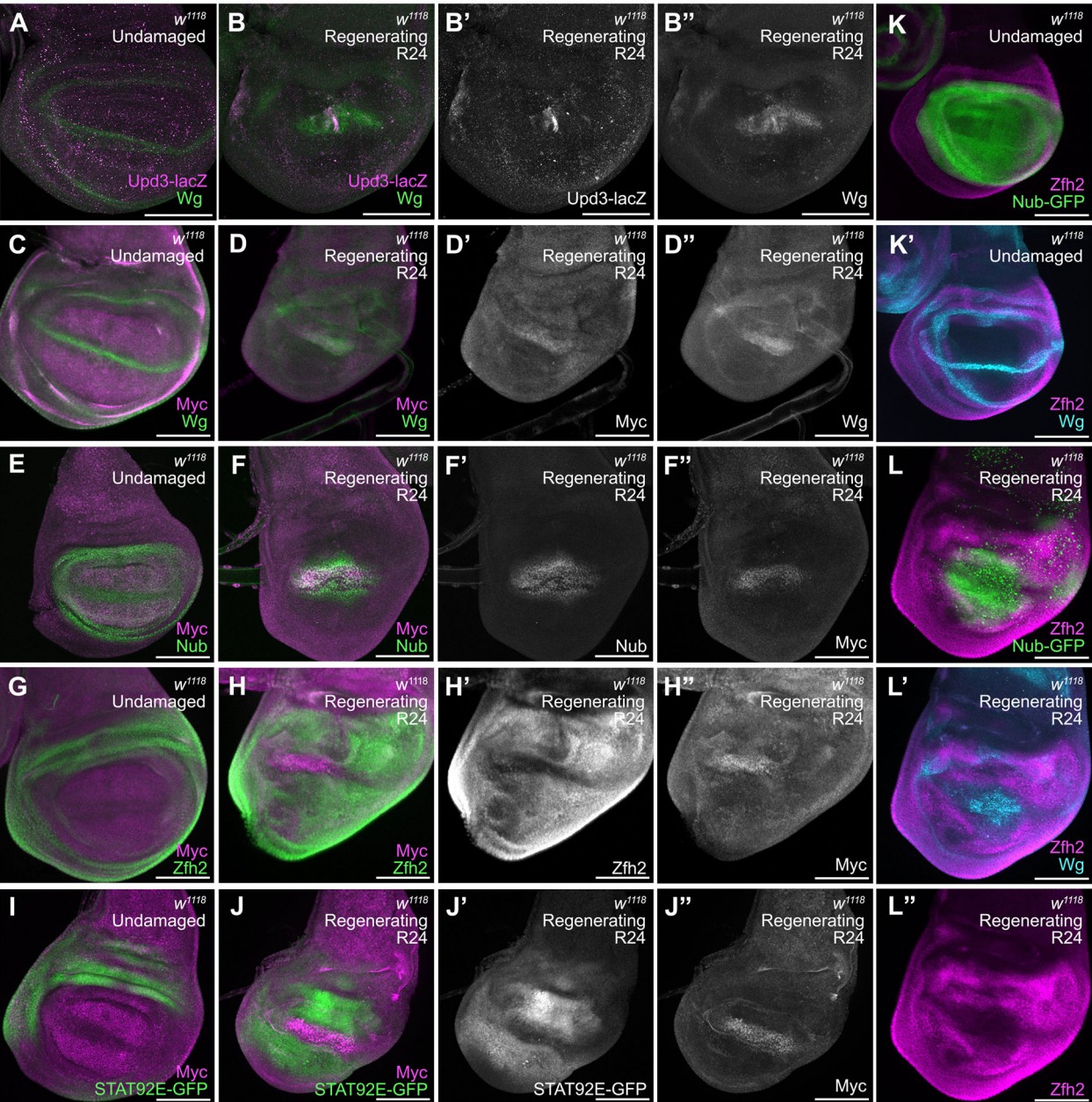

**Fig. 4. Mapping the domain of Myc expression in the regeneration blastema.** (A-B″) Wg immunostaining and *upd-lacZ* expression detected by β-galactosidase immunostaining in an undamaged disc (A) and an R24 disc (B-B″). (C-D″) Myc and Wg immunostaining in an undamaged disc (C) and an R24 disc (D-D″). (E-F″) Nub and Myc immunostaining in an undamaged disc (E) and an R24 disc (F-F″). (G-H″) Zfh2 and Myc immunostaining in an undamaged disc (G) and an R24 disc (H-H″). (I-J″) *STAT92E-GFP* expression and Myc staining in an undamaged disc (I) and an R24 disc (J-J″). (K-L″) Zfh2 and Wg staining as well as *nub-GFP* expression in an undamaged disc (K,K′) and an R24 disc (L-L″). Scale bars: 100 μm.

## The regeneration blastema includes additional zones of elevated growth outside the canonical blastema

While immunostaining for p-S6, we marked the blastema by immunostaining for Wg and found that the region with elevated Tor activity was larger than the area with Wg expression (Fig. 3A). In addition, the region marked with the OPP translation assay was also larger than the Wg-expressing area (Fig. 3J). Thus, we sought to clarify the spatial relationships among the blastema, Myc expression, Tor signaling, and enhanced translation. Given that anti-Myc and anti-p-S6 are both rabbit antibodies, and that Myc and Wg expression colocalized, we used the Wg antibody to mark the Myc-expressing blastema cells. By visualizing Wg, p-S6

and translation, we discovered additional concentric zones of the regeneration blastema. Wg, marking the canonical blastema, was expressed in the center, which we call Growth Zone A (Fig. 5A,B), and which also had elevated p-S6 and OPP signal (Fig. 5C,D). Growth Zone B, the area outside of Zone A, had high p-S6 but not Wg (Fig. 5A,C-E). Growth Zone C was the area with high translation but not upregulated Wg, Myc or p-S6 (Fig. 5A,F,G). To confirm the existence of three distinct zones, we quantified the area of each (Fig. 5H,I) and examined the fluorescence intensity profile of a cross-section through the wing disc (Fig. S5A-G′).

We wondered whether the OPP signal in Zone C that extended beyond Zone B was caused by Tor signaling non-autonomously

inducing increased translation, or by spread of the OPP signal, or by an unknown additional factor that increases translation. To test the extent to which Tor activity can induce OPP signal non-autonomously, we activated Tor in the posterior half of an undamaged wing disc using *hhGal4, UAS-Tsc2RNAi* (*hh>Tsc2i*) (Fig. S5H-Q). In these discs, the elevated OPP signal extended beyond the anterior-posterior boundary while the p-S6 immunostaining did not (Fig. S5H-Q). Therefore, either Tor activity can induce elevated translation in the adjacent cells, or the OPP assay signal can spread, perhaps due to newly synthesized secreted or extracellular proteins labeled with OPP.

To determine whether Growth Zone C coincided with the Nub-expressing area, we combined the OPP assay with Nub immunostaining (Fig. 5J-L). The Nub-positive area was slightly larger than the area with elevated translation, indicating that Growth Zone C is smaller than the remaining inner hinge (Fig. 5J-L).

## The concentric growth zones have different proliferation rates

To determine whether these blastema zones were biologically relevant, we investigated whether cell proliferation was different

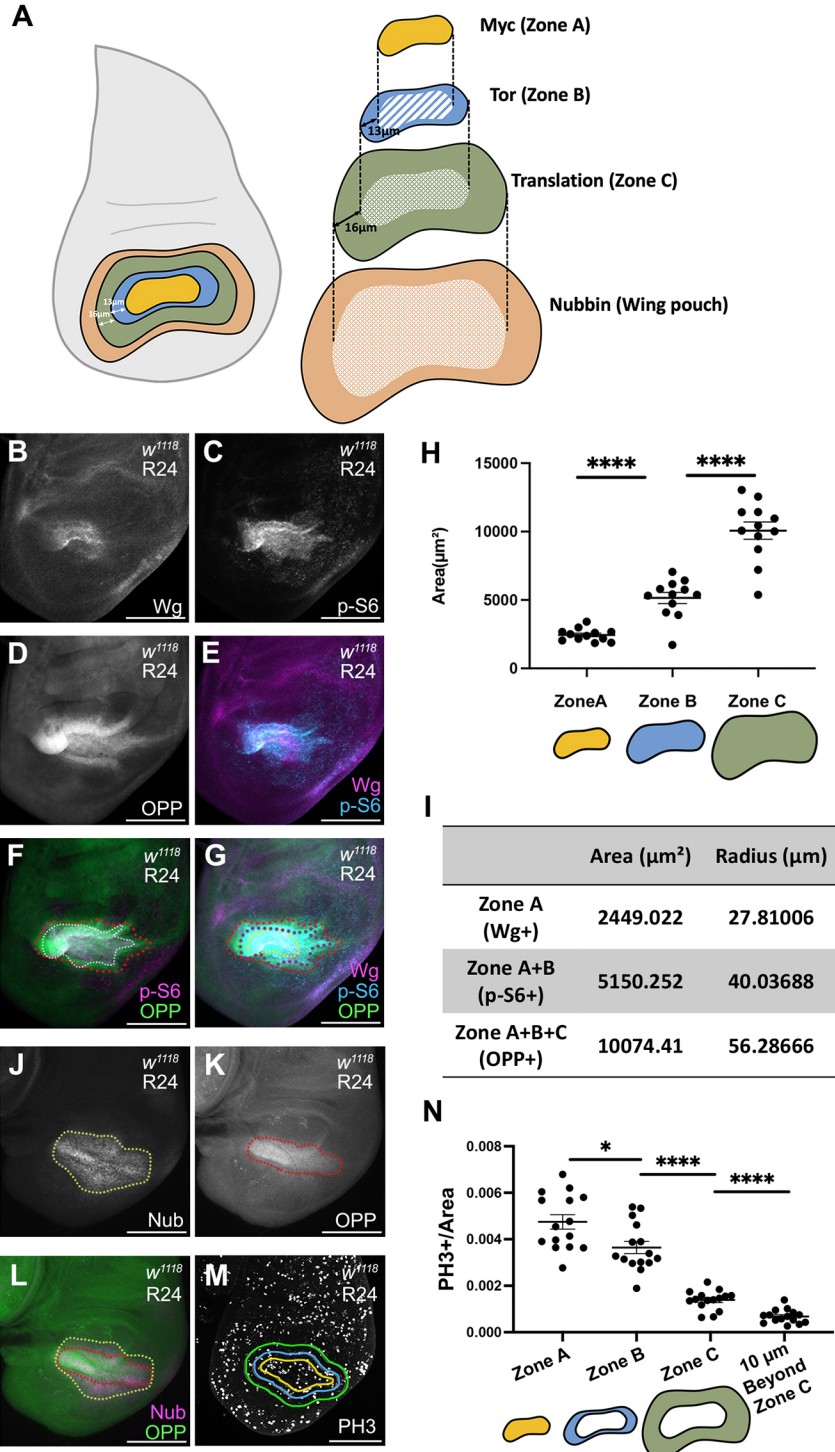

**Fig. 5. Zones of growth in the regeneration blastema.** (A) Schematic of concentric growth zones in a regenerating wing disc. (B-G) Wg staining (yellow dotted line), p-S6 staining (white dotted line) and OPP assay (red dotted line) in the same R24 disc. (B) Wg. (C) p-S6. (D) OPP assay. (E) Wg and p-S6. (F) p-S6 and OPP assay. (G) Wg, p-S6 and OPP assay. (H) Quantification of area in zones identified by Wg, p-S6 and the OPP assay. *n*=12. ****P<0.0001, Welch's *t*-test. (I) Average area and radius of each zone. (J-L) R24 disc with Nub staining (J), OPP assay (K) and merged image (L). (M) PH3 staining in an R24 disc. Yellow line outlines Growth Zone A. Blue line outlines Growth Zone B, 13 µm beyond Growth Zone A. Green line outlines Growth Zone C, 16 µm beyond Growth Zone B. (N) Quantification of mitotic cells per area in each growth zone identified as in M. *n*=15. *P<0.05, ****P<0.0001, Welch's *t*-test. Error bars are s.e.m. Scale bars: 100 µm.

across them. Given that anti-PH3 and anti-p-S6 are both rabbit antibodies, we could not use p-S6 to mark Zone B. Therefore, we took an approximate approach by measuring the area of each zone and calculating the average radius. We used the radii to determine the distances between the edges of Growth Zones A, B and C. Using these measurements, the average distance between the edges of Growth Zone A and B was 13 µm, and the average distance between the edges of Growth Zone B and C was 16 µm (Fig. 5A,I). We then immunostained for Wg to establish the boundary of Growth Zone A and measured concentric zones based on the shape of Wg immunostaining (Fig. 5M). We then assessed mitoses per area in each zone (Fig. 5N). Proliferation was highest in Growth Zone A, was slightly reduced in Growth Zone B, and was lower still in Growth Zone C. Proliferation in the cells 10 µm outside of Growth Zone C was even lower. Thus, these growth zones are biologically meaningful and likely contribute at different levels to regenerative growth.

### Differences in Myc expression induce cell death at the periphery of Growth Zone A

Given the difference in Myc levels between Growth Zone A and the rest of the disc, we wondered whether cell competition was occurring. We quantified cell death using the terminal d-UTP nick-end labeling (TUNEL) assay. Since we used apoptosis to induce tissue damage, we needed to distinguish between ablation and cell competition. We used *UAS-eGFP* to mark cells eliminated by our ablation system. At R0, GFP-expressing debris was found on both the basal and the apical sides of the epithelium (Fig. 6A). However, at R24 the GFP-containing debris on the basal side of the epithelium was largely gone (Fig. 6B), and most of the TUNEL-positive cells did not express GFP (Fig. 6B).

Cells undergoing apoptosis due to cell competition are extruded toward the basal side of the epithelium (Amoyel and Bach, 2014; Li and Baker, 2007). In addition, cell death driven by Myc-induced super-competition occurs within an eight-cell zone adjacent to the winner cells (de la Cova et al., 2004). Thus, we counted TUNEL-positive cells on the basal side of the epithelium in an eight-cell zone. To set a baseline for quantification, we quantified TUNEL in a negative control, in which *dpp-Gal4* was used to drive *UAS-GFP* in a normally developing wing disc, and a positive control, in which *dpp-Gal4* was used to drive *UAS-Myc*. In the negative control, there were no Myc expression differences and no TUNEL-positive cells (Fig. 6C,C′,G). In the positive control, cell death was observed in cells located inside the high-Myc stripe and within an eight-cell region adjacent to the high-Myc cells (Fig. 6D,D′,G), consistent with previous studies (de la Cova et al., 2004).

We then quantified TUNEL-positive cells in regenerating discs that were on the basal side of the epithelium within an eight-cell-wide band outside of the Myc-expressing cells, which measured about 39 µm (Fig. 6E,E′,H). We found that the number of TUNEL-positive cells in this eight-cell band was even higher than in the positive controls (Fig. 6E,E′,G). We examined cell death at 48 h and 72 h after tissue damage and found that this cell death continued throughout regeneration (Fig. S6A-C). To determine whether this apoptosis was due to elevated Myc expression, we quantified TUNEL-positive cells in $Myc^{P0}$ regenerating discs, which lacked Myc expression throughout the disc (Fig. 6F,F′,G, Fig. S2B). Surprisingly, the overall number of TUNEL-positive cells on the basal side of the epithelium was not reduced in $Myc^{P0}$ regenerating discs (Fig. 6G). However, the number of TUNEL-positive cells inside the Wg-positive zone in $Myc^{P0}$ discs was higher (Fig. 6H,I), and cell death at the periphery of the Wg-positive zone was

significantly reduced (Fig. 6H,J). This reduction in cell death outside of the Wg-positive zone when Myc is removed suggests that the cell death is due to Myc expression level differences. The increased cell death within the Wg-positive zone in $Myc^{P0}$ regenerating discs suggests that Myc is important for survival of these blastema cells. Given that JAK/STAT signaling can also induce cell competition (Rodrigues et al., 2012) and that JAK/STAT signaling is elevated outside the Wg-positive zone (Fig. 4J), the loss of Myc may render the cells inside the Wg-positive zone loser cells, causing the observed increase in apoptosis.

### Cell competition induced in regenerating wing discs is distinct from experimentally induced cell competition

Cell competition and super-competition studies usually involve genetic clones that differ by one or two factors relative to the surrounding cells, experimentally establishing a mismatch in cell fitness. However, in regenerating imaginal discs, there are overlapping and adjacent zones of elevated Wg, JAK/STAT, Yki, JNK, p38 and Tor signaling (Bergantiños et al., 2010; Grusche et al., 2011; Santabárbara-Ruiz et al., 2015; Smith-Bolton et al., 2009; Sun and Irvine, 2011) (Figs 4C-D′,I-J′, 5B,E), which may render fitness comparisons during regeneration more complex than the binary comparisons made in cell competition studies. Therefore, to establish whether the cells outside the high Myc zone were behaving like 'loser cells', we examined three different factors that are important for loser cell fate in cell competition and super-competition: proteotoxic stress, autophagy, and expression of Xrp1.

The distinction between cell competition and super-competition is based on whether wild-type cells are winner cells or loser cells (Morata, 2021). However, it is unclear whether the cells outside of the high-Myc zone in regenerating discs can be described as wild-type cells because Myc levels are lower than in normal discs. In addition, Myc levels in the blastema are elevated relative to the rest of the disc and relative to undamaged discs but are not as high as when Myc is overexpressed. Given that loser cells during cell competition have increased eIF2α phosphorylation (Baumgartner et al., 2021; Kiparaki et al., 2022), and winner cells during super-competition also have increased eIF2α phosphorylation (Paul et al., 2024), we examined phospho-eIF2α levels in regenerating discs to determine which type of cell competition was occurring. In an undamaged wing disc, eIF2α was phosphorylated throughout the disc (Fig. 7A). After tissue damage, phosphorylation of eIF2α was reduced in the Wg-positive zone of the regeneration blastema, which are the 'winner' cells (Fig. 7B-B″). Therefore, the enhanced translation in the blastema did not cause proteotoxic stress, and the regenerating disc may be experiencing something closer to cell competition than super-competition.

Cell elimination due to cell competition can involve an increase in autophagy (Nagata et al., 2019). Therefore, we used the autophagy marker *Atg8a-mcherry* (Hegedűs et al., 2016; Mauvezin et al., 2014) and counted mCherry-positive puncta in the cells adjacent to the Wg-positive zone (the boundary) and in the cells within the eight-cell perimeter outside of the Wg-positive zone (Fig. 7C-E, Fig. S6D,E). While there were few or no mCherry-positive puncta in undamaged wing discs (Fig. 7C, Fig. S6D), mCherry-positive puncta were observed in regenerating wing discs (Fig. 7D, Fig. S6E), particularly at the boundary and in the cells within the eight-cell perimeter outside the Wg-positive zone (Fig. 7E).

Xrp1 is a basic leucine zipper domain (bZIP) protein that is expressed in loser cells and is essential for loser cell elimination during imaginal disc cell competition (Baillon et al., 2018; Kiparaki et al., 2022; Lee et al., 2018; Ochi et al., 2021). We did not detect

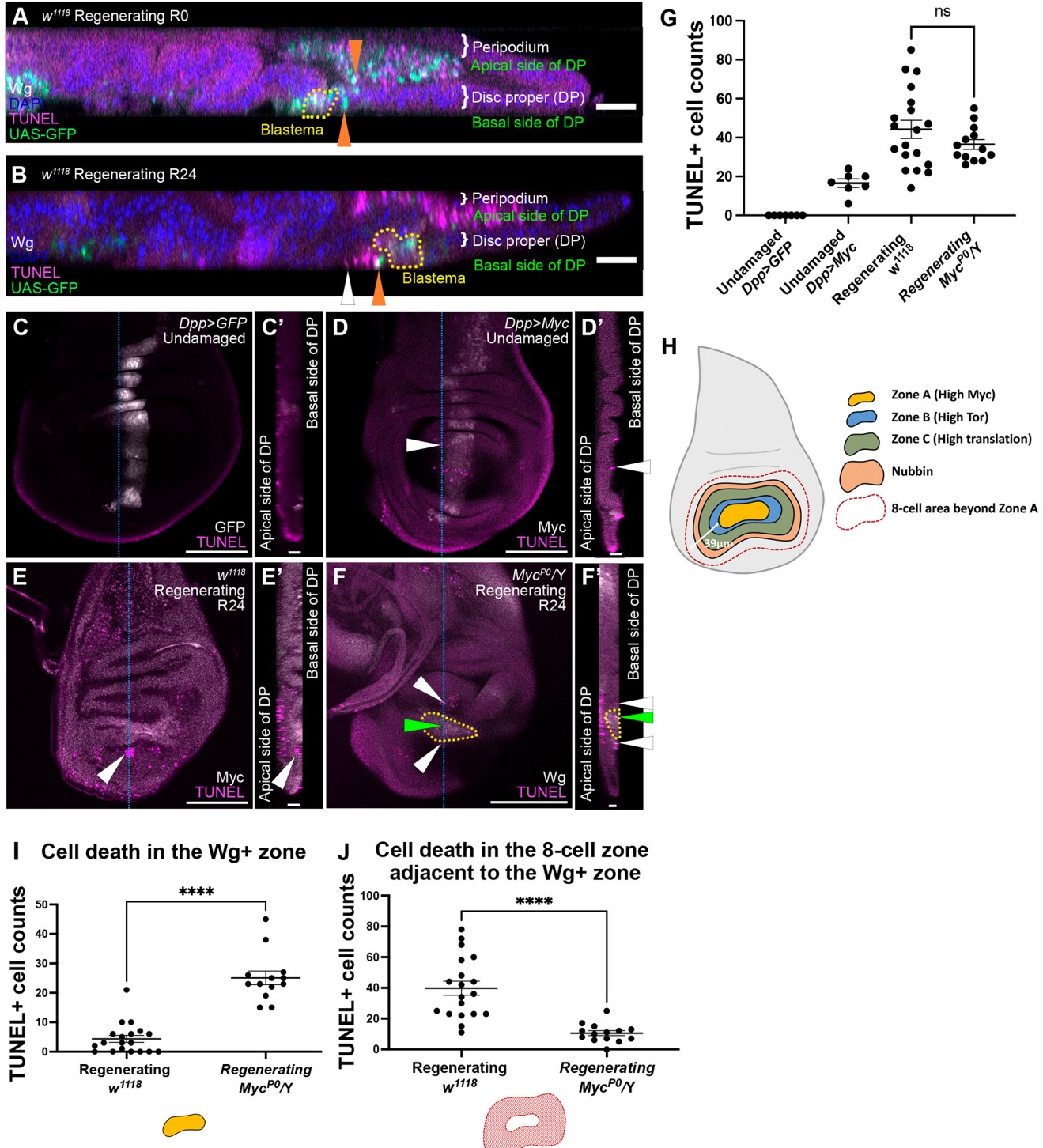

**Fig. 6. Myc expression induces cell death at the periphery of Growth Zone A.** (A,B) Orthogonal views of an R0 disc (A) and an R24 disc (B). Wg staining (white) marks the regeneration blastema (yellow dotted lines). UAS-GFP (green) marks cell debris from tissue ablation. TUNEL assay (magenta) marks debris and apoptotic cells. DAPI (blue) marks nuclei. White arrowhead marks TUNEL$^+$ GFP$^-$ apoptotic cell, orange arrowheads mark TUNEL$^+$ GFP$^+$ cell debris. (C-F′) TUNEL assay marking apoptotic cells in a negative control (*dpp-Gal4>UASGFP*) (C,C′), positive control (*dppGal4>UASMyc*) (D,D′), *w1118* R24 disc (E,E′) and *MycP0/Y* R24 disc (F,F′). (E) *w1118* R24 disc with Myc staining marking Growth Zone A. (F) *MycP0/Y* R24 disc with Wg staining (yellow dotted line) marking Growth Zone A. (C′,D′,E′,F′) Orthogonal views of C,D,E,F. White arrowheads mark apoptosis adjacent to high Myc expression. Green arrowhead marks apoptosis in the blastema. (G) Quantification of cell death labeled by TUNEL at the basal side of the epithelium in the areas that were GFP positive (C) or Myc positive (D) in the wing pouch and an eight-cell band extending anteriorly, or the Wg$^+$ Myc$^+$ blastema and an eight-cell band outside of the blastema. *dpp>GFP*, *n*=7; *dpp>Myc*, *n*=7; *w1118* R24, *n*=19; *MycP0/Y* R24, *n*=14. ns, not significant (*P*>0.05). (H) Schematic of a regenerating wing disc. Yellow indicates the Wg$^+$ Myc$^+$ zone (Growth Zone A), blue indicates Growth Zone B, green indicates Growth Zone C, red dashed line indicates the eight-cell diameter region beyond Zone A. (I) Quantification of cell death in the Wg-positive zone in R24 discs. *w1118*, *n*=19; *MycP0/Y*, *n*=14. ****P<0.0001, Welch's *t*-test. (J) Quantification of cell death in the eight-cell band outside the Wg-positive zone. *w1118*, *n*=19; *MycP0/Y*, *n*=14. ****P<0.0001. Statistical test used was Welch's *t*-test. Error bars are s.e.m. Scale bars: 100 μm.

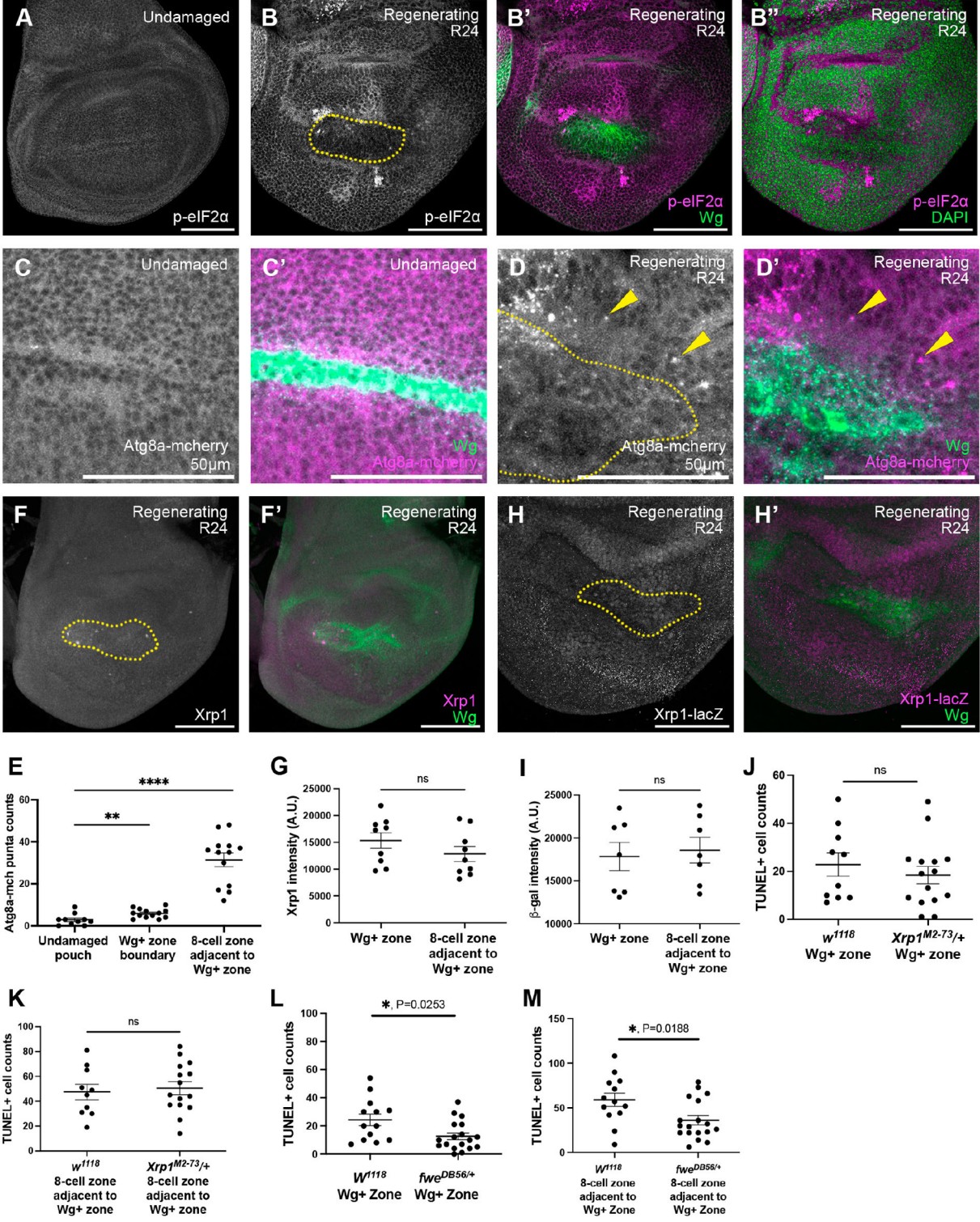

**Fig. 7. Cell competition induced in regenerating wing discs is distinct from experimentally induced cell competition.** (A,B) p-eIF2α staining in an undamaged disc (A) and an R24 disc (B). (B′) p-eIF2α and Wg in the regenerating R24 disc. (B″) p-eIF2α and DAPI in the regenerating R24 disc. (C,D) Atg8a-mcherry expression in an undamaged disc (C) and an R24 disc (D). (C′,D′) Atg8a-mcherry expression and Wg staining (yellow dotted line in D). Arrowheads indicate Atg8a puncta. Note that the images in C-D′ show a 250-pixel² area cropped from Fig. S6A,B to better show the puncta. (E) Quantification of Atg8a puncta. $n$=13. **$P$<0.01, ****$P$<0.0001. (F,F′) Xrp1 staining in an R24 disc. Yellow dotted circle indicates blastema marked by Wg. (G) Quantification of Xrp1 intensity. $n$=9. ns, not significant ($P$>0.05). (H,H′) *Xrp1-lacZ* expression in an R24 disc. Yellow dotted circle indicates blastema marked by Wg. (I) Quantification of average β-galactosidase staining intensity. $n$=7. ns, not significant ($P$>0.05). (J,K) Quantification of cell death (TUNEL) in the Wg-positive area (J) and in the eight-cell zone adjacent to the Wg-positive area (K) in $w^{1118}$ R24 discs and $Xrp1^{M2-73}$/+ R24 discs. $w^{1118}$, $n$=10; $Xrp1^{M2-73}$/+, $n$=15. ns, not significant ($P$>0.05). (L,M) Quantification of cell death (TUNEL) in the Wg-positive area (L) and in the eight-cell zone adjacent to the Wg-positive area (M) in $w^{1118}$ R24 discs and $fwe^{DB56}$/+ R24 discs. $w^{1118}$, $n$=13; $fwe^{DB56}$/+, $n$=18. Error bars are s.e.m. Statistical test used was Welch's $t$-test. A.U., arbitrary units. Scale bars: 100 μm.

expression level differences between the cells inside the blastema and the cells surrounding the blastema using an Xrp1 antibody (Brown et al., 2021) (Fig. 7F,G), an *Xrp1-GFP* in which the Xrp1 protein is tagged with GFP (Fig. S6F-F″) (Kudron et al., 2018), or an enhancer trap line *Xrp1-lacZ$^{02515}$* (Bellen et al., 2004) (Fig. 7H,I). We examined cell death in regenerating wing discs that were heterozygous for the mutant *Xrp1$^{m2-73}$*, which is a truncated allele that eliminates cell competition when heterozygous (Lee et al., 2016, 2018). Cell death surrounding the Wg-positive blastema zone was not reduced in *Xrp1$^{m2-73}$/+* discs (Fig. 7J,K, Fig. S6G,H). Thus, it is unlikely that Xrp-1 mediates this Myc-induced cell death.

Several mechanisms can mediate the fitness comparison in cell competition, including expression of two isoforms of *flower* (*fwe*), *fwe$^{loseA}$* and *fwe$^{loseB}$*, and *azot* in the loser cells (Merino et al., 2015; Rhiner et al., 2010), and expression of the Spatzle-processing enzymes SPE or ModSP in the winner cells to activate Spatzle and induce Toll signaling in loser cells (Alpar et al., 2018; Germani et al., 2018; Meyer et al., 2014). We did not find an increase in *fwe$^{loseA}$*, *fwe$^{loseB}$* or *azot* expression in regenerating wing discs in the low-Myc cells (Fig. S7A-C′,R,S). We also did not find a reduction in cell death in discs heterozygous for an *azot* mutant (Fig. S7D,R,S), and we found a global reduction in cell death in discs heterozygous mutant for the entire *fwe* locus (Fig. 7L,M, Fig. S7E,R,S). While this global reduction in cell death may indicate an overall role in cell survival, it may also indicate a role in Myc-induced competition outside Zone A and JAK/STAT-induced cell competition inside Zone A.

While SPE can be upregulated in winner cells, our published transcriptional profile suggested that *SPE* is downregulated in the blastema with a log2 decrease of 1.57 (*P*=0.00005) (Khan et al., 2017). Furthermore, whole-disc RT-qPCR showed no difference in *SPE* expression after damage (Fig. S7F). We used *in situ* hybridization chain reaction (HCR) to examine the spatial distribution of *SPE* mRNA (Fig. S7G) (Choi et al., 2018; Bruce et al., 2021). *SPE* mRNA was not different between Growth Zone A and the adjacent eight-cell band (Fig. S7H). *modSP* mRNA levels were not elevated in our previously published transcriptional profile (Khan et al., 2017). We examined spatial expression of *modSP* by HCR, which was upregulated in experimentally induced winner cells in an undamaged disc but was not upregulated in Zone A of a regenerating disc (Fig. S7I-K). To assess activation of Toll signaling, we quantified the amount of Dorsal immunostaining present in the nuclei of cells inside and outside Zone A (Fig. S7L). Given that the antibodies against Dorsal and Wg were both generated in mice, Zone A was marked by a newly generated transgenic line, *2xHA-Myc*, in which a 2xHA-tag was added to the N terminus of the endogenous *Myc* gene (Fig. S7L′). While some Dorsal translocated into the nucleus inside of the Zone A cells (Fig. S7M-N,O), its location remained cytoplasmic in the cells outside of Zone A (Fig. S7M′,N′,O). To determine whether SPE is required for death in the eight-cell zone, we quantified TUNEL in regenerating discs heterozygous for the *SPE$^{SK6}$* mutation (Fig. S7P,R,S) or expressing *UAS-SPE-RNAi* (Fig. S7Q,T,U). We did not find reduction of cell death surrounding the high-Myc area in either condition.

Taken together, these data suggest that increased Myc in the regeneration blastema leads to something closer to cell competition than super-competition, given the absence of cell death when Myc levels were equalized, the increased proteotoxic stress, and the presence of autophagy. Additionally, Myc-induced cell death outside the high-Myc zone is not identical to canonical cell competition, which is mediated by Xrp1 and can involve Azot expression or Toll signaling activation.

## DISCUSSION

We have shown that Myc and Tor are two crucial factors that regulate regenerative growth. Reduction of either factor impaired proliferation and translation in the regeneration blastema. Furthermore, we observed concentric growth zones that extend beyond the blastema as defined by transcriptional profiling. We also demonstrated that the upregulated Myc expression induces cell competition-like cell death in the surrounding cells.

The concentric growth zones were differentiated through expression of genes such as Wg and Myc, activated Tor signaling, and elevated translation. Given that the shape of each zone is similar, it is likely that some secreted factor in Growth Zone A establishes the outer zones, perhaps through damage-response signaling such as reactive oxygen species, morphogen signaling such as Wg, or mechanical signaling such as the Hippo pathway. Furthermore, the high translation in Zone C may be a non-autonomous effect of high Tor activity in Zone B. These distinct zones were not identified by scRNA-seq experiments (Floc'hlay et al., 2023; Worley et al., 2022), suggesting that the cells in Zones B and C may not have differential gene expression. However, these studies induced damage through overexpression of Eiger, whereas we induced damage through overexpression of Reaper, and the blastema after Eiger-induced ablation may not have different zones of Myc expression and Tor activity.

Evidence for a similar gradient in proliferation has been observed in zebrafish fin regeneration, where differential expression of *msx1b*, a homeodomain transcriptional repressor, gives rise to a proliferation gradient that occurs between the proximal and distal blastema (Nechiporuk and Keating, 2002). Furthermore, elevated Tor activity may be a common feature in regeneration, as Tor signaling is also essential for zebrafish fin regeneration (Hirose et al., 2014; Xiao et al., 2024), axolotl limb regeneration (Zhulyn et al., 2023) and a model of tissue inflammation and regeneration in *Drosophila* (Maya et al., 2024 preprint). Thus, transcription factors and Tor signaling may establish zones of regenerative growth in many systems.

The enhanced Myc expression in the regeneration blastema has a double-edged effect on the regenerating wing disc. While Myc drives regrowth in Growth Zone A, it also induces cell death in the neighboring cells, tempering their ability to contribute to the regenerated tissue. Although this cell death was mediated by Myc, with proteotoxic stress and autophagy increased in the loser cells, the loser cells did not upregulate Xrp1, Azot expression or Toll signaling. Thus, these interactions differed from cell competition as described in previous studies. Notably, while Growth Zone A cells had high Myc expression and thus were winner cells, the cells outside of Growth Zone A still had elevated translation. In addition, JAK/STAT signaling was also high outside of growth Zone A, and JAK/STAT signaling can both induce cell competition and rescue loser cell death (Rodrigues et al., 2012). Furthermore, JNK signaling, Yki activity and mechanical stress can also influence cell competition (Baker, 2020; Morata, 2021). Thus, regeneration occurs in a more complex signaling milieu than classical cell competition experiments and may provide clues about how cell competition operates in other complex microenvironments such as tumors. Furthermore, our work presents the first evidence of endogenous cell competition in a non-stem cell-based regeneration model, further exploration of which will enhance our understanding of both regeneration and cell competition. This exploration may include determining which genes are controlled by the transcription factor Myc and which mRNAs are experiencing enhanced translation due to the increase in ribosomal activity.

## MATERIALS AND METHODS

### Tissue ablation

Ablation experiments were performed as previously described (Smith-Bolton et al., 2009). Briefly, egg lays were performed at 25°C in the dark for 4 h, then incubated at 18°C. First-instar larvae were picked on day 2 and placed in vials (50 larvae/vial). On day 7, vials were incubated for 24 h in a 30°C circulating water bath, kept on ice for 5 min to reduce the temperature, then returned to the 18°C incubator. $Myc^{P0}$ animals had ablation induced 4 h later to coincide with the early third instar stage because they have delayed development (Abidi et al., 2023).

### Drosophila strains

Fly lines used were: $w^{1118}$ (Hazelrigg et al., 1984); $w^{1118}$; $rnGal4$, $UAS$-$rpr$, $tubGal80ts/TM6B$, $tubGal80$ (Smith-Bolton et al., 2009); $7xEcRE$-$GFP$ (Hackney et al., 2007; Terry et al., 2024) (a gift from Kenneth Moberg); $Myc$-$lacZ$ (RRID:BDSC_12247); $Myc^{P0}$ (Johnston et al., 1999) (RRID: BDSC_11298); $Tor^{2L1}$ (Oldham et al., 2000) (RRID:BDSC_98102); $Upd3$-$lacZ$ (Bunker et al., 2015) (a gift from David Bilder); $10XStat92E$-$GFP$ (Ekas et al., 2006) (RRID:BDSC_26197); Nub-MiMIC-GFP ($Mi\{MIC\}nub^{MI05126}$) (Venken et al., 2011) (RRID:BDSC_37920); $UAS$-$Myc$ (Johnston et al., 1999) (RRID:BDSC_9674); $UAS$-$GFP$ (Shiga et al., 1996) (RRID:BDSC_4775); $Dpp$-$Gal4$ (Kojima et al., 2000) (RRID: BDSC_93385); $Xrp1$-$lacZ$ (Bellen et al., 2004) (RRID:BDSC_11569); $Xrp1$-$GFP$ (Kudron et al., 2018) (RRID:BDSC_83391), $Xrp1^{M2-73}$ (Lee et al., 2018) (RRID:BDSC_81270); $3xmCherry$-$Atg8a$ (Hegedűs et al., 2016) (a gift from Juhász Gábor); $fwe^{loseA}$::$mcherry$ KI (Levayer et al., 2015); $fwe^{loseB}$::$mcherry$ KI, $azot\{KO; gfp\}$ (Merino et al., 2015) (gifts from Christa Rhiner); $fwe^{DB56}$ (RRID:BDSC_51610); $SPE^{SK6}$ (Yamamoto-Hino et al., 2015); $UAS$-$SPE$-$RNAi$ (RRID:BDSC_33926); $2xHA$-$Myc$ (this paper).

### Generation of 2xHA-Myc transgenic line

To generate an HA-Myc transgenic fly line, we used the Scarless gene editing technique (Gratz et al., 2024). Homology arms 1 kb upstream of the Myc transcription start site and 1 kb downstream of the Myc transcription start site were cloned into the pHD-2xHA-ScarlessDsRed plasmid (DGRC Stock 1366; RRID:DGRC_1366), flanking the 2xHA and the DsRed cassette (cloning performed by GenScript). The gRNA sequence (ttcagACAGGCATATAACTCAG) was cloned into the pCFD5 plasmid at the BbsI site. These plasmids were injected into PBac{y[+mDint2] GFP[E.3xP3]=vas-Cas9}VK00027 (RRID:BDSC_51324). Injection services were provided by BestGene. We crossed the 2xHA-DsRed-Myc fly lines to a piggyBac transposase line (RRID:BDSC_8285) to remove the DsRed marker.

### Immunostaining

Immunostaining was carried out as previously described (Smith-Bolton et al., 2009). After dissection, the larval carcasses were fixed in 4% paraformaldehyde (PFA) for 20 min at room temperature, followed by three washes in 0.1% Triton X-100 in PBS (0.1% PBST) for 10 min. Carcasses were incubated at 4°C overnight in primary antibodies diluted as noted below in 10% normal goat serum and 0.1% PBST, followed by three washes of 0.1% PBST for 10 min. Secondary antibody was diluted 1:1000 in 10% normal goat serum and 0.1% PBST for incubation at 4°C overnight. After the secondary antibody incubation, the samples were washed three times using 0.1% PBST, followed by incubation in 70% glycerol in PBS overnight at 4°C. The wing imaginal discs were then dissected and mounted on slides with VECTASHIELD anti-fade mounting media (Vector Laboratories, H-1000).

Primary antibodies used were: anti-Myc (1:250; Santa Cruz Biotechnology, sc-28207), anti-p-S6 (Romero-Pozuelo et al., 2017) (1:200; a gift from Aurelio Teleman), anti-phospho-histone H3 (1:500; Millipore Sigma, 06-570), anti-Nubbin (RRID:AB_2722119) [1:250; Developmental Studies Hybridoma Bank (DSHB), Nub2D4], anti-Wingless (RRID: AB_528512) (1:100; DSHB, 4D4), anti-Zfh2 (Tran et al., 2010) (1:250; a gift from Chris Doe), anti-β-galactosidase (1:500; Invitrogen, Thermo Fisher Scientific, A-11132), anti-p-eIF2α (1:100; Cell Signaling Technology, 3398), anti-Xrp1 (Brown et al., 2021) (1:1000; a gift from Hyung Don Ryoo), anti-Fibrillarin (1:100; Abcam, ab4566), anti-Dorsal (1:20; DSHB, 7A4), anti-HA (1:1000; Cell Signaling Technology, 3274). Secondary antibodies were: Alexa Fluor 488, 555, 633, 647 (1:1000; Invitrogen, A21424, A21245, A21240, A21071, A21052 and A21094) and DAPI (1:1000, Thermo Fisher Scientific, D1306).

### HCR

The HCR technique was carried out using the HCR Gold RNA-FISH kit (Molecular Instruments) as previously described (Choi et al., 2018; Bruce et al., 2021) with some modifications. After dissection, the larval carcasses were fixed in 4% PFA for 20 min at room temperature, followed by three washes in 0.1% PBST for 10 min. After the washes, the samples were incubated in pre-hybridization buffer at 37°C for 30 min, followed by incubation with probes at 37°C for two nights. Probe washing and amplification steps were conducted as described (Bruce et al., 2021). DAPI (1:1000) was added during the hairpin incubation step. Probes used were: SPE (3:200), ModSP (3:200) (Molecular Instruments).

### OPP assay

The OPP assay was conducted as described (Kiparaki and Baker, 2023), using the Click-iT Plus OPP Alexa Fluor 488 Protein Synthesis Assay Kit (Thermo Fisher Scientific, 10456). Briefly, larvae were dissected in Schneider's Drosophila Medium (Thermo Fisher Scientific, 21720024) with 10% heat-inactivated fetal bovine serum (Thermo Fisher Scientific, A3840001). Larval carcasses were incubated with component A at the concentration of 10 µM in Schneider's Drosophila Medium with 10% heat-inactivated fetal bovine serum for 15 min. After rinsing with PBS, the carcasses were fixed in 4% PFA for 20 min, followed by three washes for 10 min each with 0.1% PBST. After washing, carcasses were blocked in 3% bovine serum albumin (BSA) in PBS for 10 min before the 30 min incubation in the cocktail solution for the click-on reaction. After the reaction, samples were washed with the rinse buffer followed by 2 min of blocking in 3% BSA in PBS. Protocols after this step followed the general primary and secondary staining method above.

### EdU incorporation and detection

EdU incorporation and detection was carried out as previously described (Gouge and Christensen, 2010) using the Click-iT EdU kit (Invitrogen, C10338). Briefly, larvae were dissected in PBS and incubated in EdU at the concentration of 100 µM in Schneider's Drosophila Medium for 30 min at room temperature, followed by fixation in 4% PFA for 20 min, followed by three washes of 10 min each with 0.1% PBST. After washing, carcasses were blocked in 3% BSA in PBS for 30 min before the 30-min incubation in the cocktail solution for the click-on reaction. Protocols after this step followed the general primary and secondary staining method above.

### TUNEL assay

The TUNEL assay was carried out using the In Situ Cell Death Detection Kit (Roche, 12156792910). For experiments incorporating the TUNEL assay, we followed the general immunostaining protocol described above. However, for the secondary antibody solution, we used the TUNEL reaction mixture made by mixing the Enzyme Solution and Label Solution at a 1:9 ratio. Secondary antibodies and DAPI were added according to the dilutions listed above. Samples were incubated in this mixture at 4°C overnight, followed by the remainder of the protocol for general immunostaining.

### Rapamycin food

Rapamycin (Thermo Fisher Scientific, AAJ62473MC) was dissolved in ethanol at a 50 mM concentration. The rapamycin solution was added to Nutri-Fly Bloomington food (Genesee Scientific, 66-121) to make a 20 µM working concentration. For control food, the same amount of ethanol was added. Immediately after the temperature shift in the circulating water bath, larvae were transferred to rapamycin food or control food.

### qPCR

qPCR was carried out as previously described (Abidi et al., 2023; Bose et al., 2025). Briefly, 40-50 wing imaginal discs were collected in Schneider's medium (Sigma-Aldrich, S1046), followed by RNA extraction using the RNeasy Kit (QIAGEN, 74104). cDNA synthesis was performed using the SuperScript III First Strand Synthesis Kit (Thermo Fisher Scientific, 11752-050). qPCR reactions were carried out using Power SYBR Green

MasterMix (Thermo Fisher Scientific, A25742) on an ABI StepOnePlus Real-Time PCR System. Primers used were: *Gapdh2* forward primer, GTGAAGCTGATCTCTTGGTACGAC; *Gapdh2* reverse primer , CCGCGCCCTAATCTTTAACTTTTAC (Classen et al., 2009); *SPE* forward primer, ACCAATACGACCCTCTGGGA; *SPE* reverse primer, GCAGTCAGGATCGGTACGAG (Alpar et al., 2018).

## Image acquisition
Discs were imaged on a Zeiss LSM 700, Zeiss LSM 880 or Zeiss LSM 900 confocal microscope. All images for a given experiment were acquired on the same microscope with the same instrument settings. Images were processed using ImageJ (NIH). Maximum intensity projections were created for the confocal images. Unless otherwise mentioned, the area used for quantification of fluorescence was determined by co-staining with Wg as a Zone A marker. The areas used for quantification for the EdU assay and the *Myc-lacZ* experiments were determined by morphology. The areas used for quantification of Dorsal in the nucleus was determined by expression of 2xHA-Myc. Within the area expressing 2xHA-Myc, DAPI staining was used to select five nuclei per disc.

## Nucleolus measurements
Ten nucleoli (anti-Fibrillarin staining) were measured in a 200×200-pixel square using ImageJ.

## Intensity plots
Intensity plots were made using ImageJ. Briefly, a line across the area of interest was selected using the line selection tool, then an intensity plot was generated using the 'Plot Profile' tool.

## Statistical analysis
All statistical analyses were performed using Welch's *t*-test on GraphPad Prism. Graphs were generated using GraphPad Prism.

### Acknowledgements
We thank Anish Bose, Snigdha Mathure and Connor Powers for critical reading of the manuscript and helpful discussions. We thank Dr Kenneth Moberg, Dr David Bilder, Dr Chris Doe, Dr Hyung Don Ryoo, Dr Aurelio Teleman, Dr Juhász Gábor, Dr Christa Rhiner and Dr Satoshi Goto for reagents; the Bloomington *Drosophila* Stock Center (NIH P40OD018537); the Developmental Studies Hybridoma Bank (NICHD, The University of Iowa); and Dr Glenn Fried, Dr Duncan Nall and Dr Umnia Doha of the Core Facilities Microscopy suite at the Carl R. Woese Institute for Genomic Biology for assistance with microscopy.

### Competing interests
The authors declare no competing or financial interests.

### Author contributions
Conceptualization: F.T.-Y.H., R.K.S.-B.; Funding acquisition: R.K.S.-B.; Investigation: F.T.-Y.H.; Writing – original draft: F.T.-Y.H., R.K.S.-B.; Writing – review & editing: R.K.S.-B.

### Funding
This work was supported by the National Institutes of Health (R01GM107140 and R35GM141741 to R.K.S.-B.). Open Access funding provided by University of Illinois Urbana-Champaign. Deposited in PMC for immediate release.

### Data and resource availability
Raw data will be posted in the Illinois Databank (https://doi.org/10.13012/B2IDB-8145148_V1). All other relevant data and details of resources can be found within the article and its supplementary information.

### Peer review history
The peer review history is available online at https://journals.biologists.com/dev/lookup/doi/10.1242/dev.204760.reviewer-comments.pdf

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
