## [Peer Review File · Development (Cambridge, England)]

Myc and Tor drive growth and cell competition in the regeneration blastema of *Drosophila* wing imaginal discs

Felicity Ting-Yu Hsu and Rachel K. Smith-Bolton

DOI: 10.1242/dev.204760

Editor: Kenneth Poss

Review timeline

Original submission:	28 February 2025
Editorial decision:	8 April 2025
First revision received:	13 September 2025
Editorial decision:	19 October 2025
Second revision received:	27 October 2025
Accepted:	29 October 2025

Original submission

First decision letter

MS ID#: dev.204760

MS TITLE: Myc and Tor drive growth and cell competition in the regeneration blastema of *Drosophila* wing imaginal discs

AUTHORS: Felicity Ting-Yu Hsu and Rachel K. Smith-Bolton

Dear Rachel,

I have now received all the referees' reports on the above manuscript, and have reached a decision. The referees' comments are appended below, or you can access them online: please go to:

As you will see, the referees express considerable interest in your work, but have some significant criticisms and recommend a substantial revision of your manuscript before we can consider publication. If you are able to revise the manuscript along the lines suggested, which may involve further experiments, I will be happy receive a revised version of the manuscript. Your revised paper will be re-reviewed by one or more of the original referees, and acceptance of your manuscript will depend on your addressing satisfactorily the reviewers' major concerns. Please also note that Development will normally permit only one round of major revision. If it would be helpful, you are welcome to contact us to discuss your revision in greater detail. Please send us a point-by-point response indicating your plans for addressing the referees' comments, and we will look over this and provide further guidance.

Please attend to all of the reviewers' comments and ensure that you clearly highlight all changes made in the revised manuscript. Please avoid using 'Tracked changes' in Word files as these are lost in PDF conversion. I should be grateful if you would also provide a point-by-point response detailing how you have dealt with the points raised by the reviewers in the 'Response to Reviewers' box. If you do not agree with any of their criticisms or suggestions please explain clearly why this is so.

Reviewer 1

Advance summary and potential significance to field

This study uses the *Drosophila* wing imaginal disc to elucidate genetic and cellular mechanisms of tissue regeneration, in particular how signals within and outside the blastema drive growth. The authors find that both Myc and Tor crosstalk to regulate cell proliferation, protein translation and ribosome biogenesis using a variety of cell biological, genetic, and pharmacological assays. In doing so, they discover distinct zones within the blastema that they characterize in terms of their rates of cell proliferation, death, Myc and Tor expression, as well as protein translation. Interestingly, the zones appear to evoke a similar paradigm to cell competition, where in the cell expressing Myc outcompetes and induce enhanced apoptosis in neighboring cells. Overall, this is a robust and rigorous study revealing a deeper understanding of how growth signals are coordinated to regulate regeneration, which are likely conserved in other models.

Comments for the author

Major Comments:

1. This study is based on identification of the blastema, but it is unclear how the authors identify this cellular structure in their images. Are the discs co-stained with a marker for these cells or is it based on morphology? This should be described in the results and methods and for consistency a dashed line should outline the blastema in all images shown.
2. Line 384 states that the "growth zones are biologically meaningful and contribute at different levels to the regrown wing pouch." Yet, the data presented in Figure 5 does not show this... it can only be inferred from the mitotic marker (P-PH3). Can the authors demonstrate that the zones do in fact contribute to different levels of growth in the wing pouch by lineage labeling or genetically eliminating a zone to demonstrate that the wing does not develop normally?
3. It would be helpful to provide the cell death characterization in terms of zone area, analogous to Figure 5N. It is unclear how the 8-cell zone of apoptosis (Tunel+) in Figure 6 relates to the zone areas described in Figure 5 (Zones: A, B, C, and wing pouch).

Minor Comments:

1. The data showing crosstalk between Myc and Tor is an important finding and should be shown in a main figure. It seems that Figure 2 and 3 can be consolidated as they show the same results with Myc and Tor and seeing them side by side would be helpful to further demonstrate the crosstalk between these genes.
2. Figure 7D. This is the only image that does not show the entire disc and it looks like there is high Atg8a-mcherry signal within the blastema on edge where the image is cut off. The entire disc should be shown with a zoom in to denote puncta measured.
3. Figure S3 Q: In the legend, this is described as a quantification of O-P' (p-S6 staining in MycP0/Y), but the axis titles say "OPP intensity" and "Tor2L1/+"
4. Figure S2: C and D are not cited and not directly related to the other images in this figure. Also, Figure S4 is not referenced in the text.
5. Describe/define the difference between cell competition and super-competition for readers unfamiliar with these terms.
6. Show representative images for graph in Figure 7J and 7K.

Reviewer 2*Advance summary and potential significance to field*

This manuscript describes the spatial relationship between the expression patterns of factors that are important for regeneration Wg and Myc with the distribution of phosphorylated S6 and OPP-detected protein translation, which they use as evidence that the blastema is partitioned into zones. The authors subsequently relate myc expression to their cell death observations as cell-cell competition between the zone expressing Wg/Myc and the surrounding cells in the adjacent zone that lack Wg/Myc expression.

The messages of this manuscript are the characterization of wing disc regeneration into zones based on the expression patterns of wg, myc, cell proliferation phosphorylated S6 and OPP-

detected protein translation, and the possible relation of myc expression with cell-cell competition. In my opinion, the novelty in the manuscript can come from the potential for describing how cell-cell competition and cell fitness can be involved in structuring the blastema and in making sure everything is regenerated with the correct organization. I believe that more characterization of cell-cell competition and how this competition is important is necessary.

The weaknesses of the manuscript: Myc has already been shown to be important in *Drosophila* wing discs (Smith-Bolton et al., *Dev. Cell* 10.1016/j.devcel.2009.04.015). One can also expect that pS6 activity (Tor activity) will be important, because of the known importance to protein translation (Zhulyn et al., *Nature* 10.1038/s41586-023-06365-1), cell proliferation (Hirose et al., *BMC Dev. Biol.* 14: 42, 2014) to regeneration. In addition, previous findings show the importance of Tor function in the size control of wing discs (Parker and Struhl, *PLoS Biol.* 10.1371/journal.pbio.1002274). The concept that blastemas consist of different regions/zone based on gene expression and cell proliferation has also been established (Poleo et al., *Dev. Dyn.* 10.1002/dvdy.1152, 2001; Santos-Ruiz et al., *Dev. Dyn.* 10.1002/dvdy.10055, 2002; Nechiporuk and Keating, *Development* 10.1242/dev.129.11.2607, 2002). In my opinion, the direction that the authors pursue to link cell competition between regions within the blastema can lead to a meaningful advancement with more evidence for the competition and the importance of the competition.

Comments for the author

I feel the evidence the authors provide of cell-cell competition is possible but needs additional support. Apoptosis is good evidence, but it can also occur when cells are not receiving any growth factor information or encountered an unsurmountable internal issue that does not involve a competition between two viable cells. Similarly, S6 phosphorylation can occur in viable cells to pause transcription. There are several types of experiments that would provide additional support.

As a few suggestions: Azot is a gene that is specifically transcribed by losing cells, and its removal (such as with RNAi) resulted in survival of the loser cells (Merino et al., *Cell*: 10.1016/j.cell.2014.12.017). Flower is another factor that determines loser status. Winner cells express Flowerubi, while loser cells will express one of two loser isoforms Flowerlose (Rhiner et al., *Dev. Cell* 10.1016/j.devcel.2010.05.010). The authors. Both factors were identified in *Drosophila* and characterized using the wing disc, so the authors can provide additional evidence for their hypothesis about cell competition between Myc/Wg expressing and surrounding cells by detecting the expression of azot and one of the flowerlose isoforms.

To provide evidence that the competition is genuinely important and why, the authors can test whether elimination of these genes results in increased survival of the loser cells, and they can find out how important the described cell competition phenomenon is for regeneration in the wing disc based on how well regeneration of the disc occurs and how it affects the subsequent the formation of adult wing, since these factors have been shown to be important for cell-cell competition relationships that affect wing disc's development (Rhiner et al., *Dev. Cell* 10.1016/j.devcel.2010.05.010; Merino et al., *Cell* 10.1016/j.cell.2014.12.017) and altered development subsequently perturbs the form of the adult wing (Merino et al., *Cell* 10.1016/j.cell.2014.12.017). If temporally disrupting Azot or Flower during the regeneration period alters the regeneration of the disc and possibly the formation of the adult wing, then this will provide much stronger evidence that cell competition is involved as the authors postulate and show how important cell competition is for the function of the blastema.

Minor points:

The section title for figure 6 (line 386) is not what the authors tested, since their experiment tested the outcome of the reduction in Myc levels with the hypomorph. A more accurate title should be along the lines of the reduction of Myc increases cell death in Wg-expressing cells and decreases in adjacent Wg-negative cells.

Reviewer 3

Advance summary and potential significance to field

This study seeks to define the growth regulatory roles of Myc and Tor in a fly wing disc regeneration model that utilizes a Gal4/Gal80 temperature-controlled reaper transgene. The translation (OP-puromycin), EdU, and pH3 experiments in the first half of the figures provide a series of readouts for the effect of Myc and Tor alleles in the second part of the paper. The experiments confirm that reducing the genetic dose of Myc or Tor individually impair local proliferation and growth. Comparing readouts of each pathway indicate that Myc and Tor pathways exhibit reciprocal dependence on the other, although the mechanism(s) of this crosstalk is not investigated. The authors also address the interesting question of whether blastema cells with elevated Myc use cell competition to eliminate adjacent cells at the border of the pouch. Establishing a physiologic role for cell competition during regeneration would be a significant advance. The data indicate that Myc expressing blastema cells are able to eliminate nearby "Myc-low" cells, but do not progress to testing whether this competitive apoptosis is required for efficient wing regrowth or patterning.

As it stands, the paper relies on genetics to infer relationships between Myc-Tor and between competition and regeneration. Tests of each of these linkages would enhance mechanistic insight.

- 1) Myc promotes expression of ribosomal subunit proteins and RNAs. Tor promotes ribosome assembly by activating S6/S6k. Do alleles of these genes synergistically impair regrowth and, at a protein level, do Myc and Tor collaborate to produce and assemble more ribosomes in blastema cells?
- 2) Is cell competition required for efficient or accurate regeneration? The data show that competition occurs, but it could be a simple clearance mechanism to remove damaged cells rather than a selection process to ensure regrowth by the fittest cells. Blocking competition should reveal what role it plays in wing regeneration.

Minor point:

1. The authors need to be careful in drawing comparisons between gene expression in the Reaper regeneration systems used in this study and the Eiger regeneration system used in the Worley and Floc'hlay single cell seq studies. Reaper is a cleaner death signal that acts immediately above caspases, while Eiger triggers Hid/Reaper but also activates JNK, which primes regeneration by activating AP1 transcriptional activity. Regeneration after Eiger-death exhibits some differences with Reaper, including a stronger ecdysone response. It's ok to make comparisons between the Reaper and Eiger systems, but there are caveats that should be mentioned.

First revision

Author response to reviewers' comments

We thank the reviewers for their thoughtful comments and suggestions, and have addressed each question to the extent that was technically possible.

Reviewer 1: SUMMARY OF THE ADVANCE MADE IN THIS PAPER AND ITS POTENTIAL SIGNIFICANCE TO THE FIELD

This study uses the *Drosophila* wing imaginal disc to elucidate genetic and cellular mechanisms of tissue regeneration, in particular how signals within and outside the blastema drive growth. The authors find that both Myc and Tor crosstalk to regulate cell proliferation, protein translation and ribosome biogenesis using a variety of cell biological, genetic, and pharmacological assays. In doing so, they discover distinct zones within the blastema that they characterize in terms of their rates of cell proliferation, death, Myc and Tor expression, as well as protein translation. Interestingly, the zones appear to evoke a similar paradigm to cell competition, where in the cell expressing Myc outcompetes and induce enhanced apoptosis in neighboring cells. Overall, this is a robust and rigorous study revealing a deeper understanding of how growth signals are coordinated to regulate regeneration, which are

likely conserved in other models.

SUGGESTIONS TO AUTHORS

Major Comments:

1. This study is based on identification of the blastema, but it is unclear how the authors identify this cellular structure in their images. Are the discs co-stained with a marker for these cells or is it based on morphology? This should be described in the results and methods and for consistency a dashed line should outline the blastema in all images shown.

We have added a dashed outline to the images.

We have clarified how the blastema was marked in the material and methods sections (Lines # 713-719).

“If not otherwise mentioned, the area used for quantification of fluorescence was determined by co-staining for Wg as a Zone A marker (yellow dotted lines). The areas used for quantification for the EdU assay and the *Myc-lacZ* experiments were determined by morphology. The areas used for quantification of Dorsal in the nucleus was determined by expression of 2xHA-Myc. Within the area expressing 2xHA-Myc, DAPI staining was used to select five nuclei per disc.”

2. Line 384 states that the "growth zones are biologically meaningful and contribute at different levels to the regrown wing pouch." Yet, the data presented in Figure 5 does not show this... it can only be inferred from the mitotic marker (P-PH3). Can the authors demonstrate that the zones do in fact contribute to different levels of growth in the wing pouch by lineage labeling or genetically eliminating a zone to demonstrate that the wing does not develop normally?

We have rewritten the text to clarify that proliferation levels are different (lines 386-387). “Thus, these growth zones are biologically meaningful and likely contribute at different levels to regenerative growth.”

We attempted to use the L-trace (G-trace with LexA) system to trace the cells outside Zone A, using *GMR26E03-lexA* as a spatially restricted driver as described in Worley et al., 2022, to quantify their contribution to the regenerated pouch. However, in our hands *GMR26E03-lexA* drives expression only in some of the inner hinge cells and also drives expression in a substantial portion of the pouch, using *Wingless* expression as a landmark to denote the location of the hinge and pouch. Thus, without a suitable LexA driver restricted to the hinge, the lineage tracing was not informative.

Below: Expression pattern of *GMR26E03-LexA* in undamaged and regenerating discs shows pouch expression (arrows), indicating that this driver does not just label hinge cells or cells outside of Zone A. The L-trace experiment shows that even in the absence of damage, a significant portion of the pouch would be labeled.

3. It would be helpful to provide the cell death characterization in terms of zone area,

analogous to Figure 5N. It is unclear how the 8-cell zone of apoptosis (Tunel+) in Figure 6 relates to the zone areas described in Figure 5 (Zones: A, B, C, and wing pouch).

We added a diagram that includes the 8-cell zone and growth zones to show their relationship (Fig6H), and added the average diameter of the 8-cell zone to the text (lines 413-415).

“We then quantified TUNEL-positive cells in regenerating discs that were on the basal side of the epithelium within an eight-cell-wide band outside of the Myc-expressing cells, **which measured about 39 μm** (Fig. 6E,E',H).”

Minor Comments:

1. The data showing crosstalk between Myc and Tor is an important finding and should be shown in a main figure.

We added the crosstalk data to Fig3, panels T-Y.

It seems that Figure 2 and 3 can be consolidated as they show the same results with Myc and Tor and seeing them side by side would be helpful to further demonstrate the crosstalk between these genes.

We understand the reviewer’s point of view. However, given that Figure 2 has V panels and Figure 3 now has Y panels with the addition of the crosstalk data, a combined figure would be too large. Hence, we propose to keep the figures separate.

2. Figure 7D. This is the only image that does not show the entire disc and it looks like there is high Atg8a-mcherry signal within the blastema on edge where the image is cut off. The entire disc should be shown with a zoom in to denote puncta measured.

We have added Atg8a images without cropping as FigS6 A,B.

3. Figure S3 Q: In the legend, this is described as a quantification of O-P' (p-S6 staining in Myc^{P0}/Y), but the axis titles say "OPP intensity" and "Tor2L1/+"

We appreciate the reviewer for pointing this out. The correct graph is now included as Figure 3V.

4. Figure S2: C and D are not cited and not directly related to the other images in this figure.

This content was inadvertently removed when we were editing for length. We have restored the original text (Lines 213-214).

“Similarly, measurement of mitoses by anti-PH3 staining showed fewer mitotic cells in the Myc^{P0} blastema (Fig. 2D-F, **S2C-D**).”

Also, Figure S4 is not referenced in the text.

This content was inadvertently removed when we were editing for length. We have restored the original text (Lines 328-329).

“In regenerating wing discs, we also found Nub-GFP expression overlapping with Zfh2 expression (Fig. 4L, **S4**).”

5. Describe/define the difference between cell competition and super-competition for readers unfamiliar with these terms.

We have added additional explanation to the introduction (Lines 89-92).

“**More specifically, cell competition occurs when wild-type cells are winners and cells with reduced Myc expression are losers, and super-competition occurs when cells with elevated Myc expression are winners and wild-type cells are losers** (Reviewed in Morata, 2021).”

6. Show representative images for graph in Figure 7J and 7K.

We have added these images to Figure S6.

Reviewer 2: SUMMARY OF THE ADVANCE MADE IN THIS PAPER AND ITS POTENTIAL SIGNIFICANCE TO THE FIELD

This manuscript describes the spatial relationship between the expression patterns of factors that are important for regeneration *Wg* and *Myc* with the distribution of phosphorylated S6 and OPP-detected protein translation, which they use as evidence that the blastema is partitioned into zones. The authors subsequently relate *myc* expression to their cell death observations as cell-cell competition between the zone expressing *Wg/Myc* and the surrounding cells in the adjacent zone that lack *Wg/Myc* expression.

The messages of this manuscript are the characterization of wing disc regeneration into zones based on the expression patterns of *wg*, *myc*, cell proliferation phosphorylated S6 and OPP-detected protein translation, and the possible relation of *myc* expression with cell-cell competition. In my opinion, the novelty in the manuscript can come from the potential for describing how cell-cell competition and cell fitness can be involved in structuring the blastema and in making sure everything is regenerated with the correct organization. I believe that more characterization of cell-cell competition and how this competition is important is necessary.

The weaknesses of the manuscript: *Myc* has already been shown to be important in *Drosophila* wing discs (Smith-Bolton et al., *Dev. Cell* 10.1016/j.devcel.2009.04.015). One can also expect that pS6 activity (Tor activity) will be important, because of the known importance to protein translation (Zhulyn et al., *Nature* 10.1038/s41586-023-06365-1), cell proliferation (Hirose et al., *BMC Dev. Biol.* 14: 42, 2014) to regeneration. In addition, previous findings show the importance of Tor function in the size control of wing discs (Parker and Struhl, *PLoS Biol.* 10.1371/journal.pbio.1002274). The concept that blastemas consist of different regions/zone based on gene expression and cell proliferation has also been established (Poleo et al., *Dev. Dyn.* 10.1002/dvdy.1152, 2001; Santos-Ruiz et al., *Dev. Dyn.* 10.1002/dvdy.10055, 2002; Nechiporuk and Keating, *Development* 10.1242/dev.129.11.2607, 2002). In my opinion, the direction that the authors pursue to link cell competition between regions within the blastema can lead to a meaningful advancement with more evidence for the competition and the importance of the competition.

We have added relevant references to the discussion (Lines 551-557).

“Evidence for a similar gradient in proliferation has been observed in other regeneration models. For example, in zebrafish fin regeneration, differential expression of *msxb*, a homeodomain transcriptional repressor, gives rise to a proliferation gradient that occurs between the proximal and distal blastema (Nechiporuk and Keating, 2002). In addition, mTor signaling is also essential for zebrafish fin regeneration (Hirose et al., 2014; Xiao et al., 2024) and axolotl limb regeneration (Zhulyn et al., 2023). Thus, transcription factors and Tor signaling may establish zones of regenerative growth in many systems.”

SUGGESTIONS TO AUTHORS

I feel the evidence the authors provide of cell-cell competition is possible but needs additional support. Apoptosis is good evidence, but it can also occur when cells are not receiving any growth factor information or encountered an unsurmountable internal issue that does not involve a competition between two viable cells. Similarly, S6 phosphorylation can occur in viable cells to pause transcription. There are several types of experiments that would provide additional support.

The apoptosis depends on a difference in *Myc* levels, and is lost when *Myc* levels are equalized, supporting the comparison to cell competition.

As a few suggestions: *Azot* is a gene that is specifically transcribed by losing cells, and its

removal (such as with RNAi) resulted in survival of the loser cells (Merino et al., Cell: 10.1016/j.cell.2014.12.017). Flower is another factor that determines loser status. Winner cells express Flowerubi, while loser cells will express one of two loser isoforms Flowerlose (Rhiner et al., Dev. Cell 10.1016/j.devcel.2010.05.010). The authors. Both factors were identified in *Drosophila* and characterized using the wing disc, so the authors can provide additional evidence for their hypothesis about cell competition between Myc/Wg expressing and surrounding cells by detecting the expression of azot and one of the flowerlose isoforms.

To provide evidence that the competition is genuinely important and why, the authors can test whether elimination of these genes results in increased survival of the loser cells, and they can find out how important the described cell competition phenomenon is for regeneration in the wing disc based on how well regeneration of the disc occurs and how it affects the subsequent the formation of adult wing, since these factors have been shown to be important for cell-cell competition relationships that affect wing disc's development (Rhiner et al., Dev. Cell 10.1016/j.devcel.2010.05.010; Merino et al., Cell 10.1016/j.cell.2014.12.017) and altered development subsequently perturbs the form of the adult wing (Merino et al., Cell 10.1016/j.cell.2014.12.017). If temporally disrupting Azot or Flower during the regeneration period alters the regeneration of the disc and possibly the formation of the adult wing, then this will provide much stronger evidence that cell competition is involved as the authors postulate and show how important cell competition is for the function of the blastema.

The Azot/Flower system and Spatzle/Toll signaling have both been shown to mediate fitness comparisons in different contexts. To assess the extent to which they are deployed in regenerating wing discs, we examined expression of FwelopeA, FwelopeB and Azot. We also examined expression of SPE, modSP, and nuclear levels of Dorsal (as a readout of Toll signaling). However, quantification showed no elevation of fwelosaA, fweloseB, or azot in the 8-cell band, no elevation of SPE or modSP in the high-Myc cells, and no increase in Toll signaling in the 8-cell band (Fig S7, A-O). These findings are consistent with our conclusion that the cell competition in regeneration is not identical to experimentally induced cell competition.

We quantified cell death in *fwe*^{DB56/+}, *Azot*^{KO/+}, *SPE*^{sk6/+}, and *SPE*-RNAi discs. Cell death in the “loser cell” zone was not reduced in *Azot*^{KO/+}, *SPE*^{sk6/+}, and *SPE*-RNAi discs (Figure S7P-U). Interestingly, *fwe*^{DB56} heterozygotes showed an overall reduction in cell death. However, given that cell death was also reduced in the blastema, we are unsure if this reduction in cell death in the loser cell zone is due to removal of cell competition or overall reduction of apoptosis.

Lines 488-526

“Several mechanisms can mediate the fitness comparison in traditional cell competition, including expression of the loser isoforms of *flower* (*fwe*), *fwe*^{loseA} and *fwe*^{loseB}, and *azot* in the loser cells (Merino et al., 2015; Rhiner et al., 2010), and expression of the Spatzle-processing enzymes SPE or ModSP in the winner cells to activate Spatzle and induce Toll signaling in loser cells (Alpar et al., 2018; Germani et al., 2018; Meyer et al., 2014). We did not find an increase in *fwe*^{loseA}, *fwe*^{loseB}, or *azot* expression in regenerating wing discs in the low-Myc cells (Fig. S7A-C', R-S). We also did not find a reduction in cell death in discs heterozygous for an *azot* mutant (Fig. S7D, R-S), and we found a global reduction in cell death in discs heterozygous mutant for the entire *fwe* locus (Fig. S7E, R-S).

While SPE can be upregulated in winner cells, our published transcriptional profile suggested that SPE is downregulated in the blastema with a log2 decrease of 1.57 (p=0.00005) (Khan et al., 2017). Furthermore, whole-disc RT-qPCR showed no difference in SPE expression after damage (Fig. S7F). We used in situ hybridization chain reaction (HCR) to examine the spatial distribution of SPE mRNA (Fig. S7G) (Choi et al., 2018; S Bruce et al., 2021). SPE mRNA was not different between Growth Zone A and the adjacent 8-cell band (Fig. S7H). modSP mRNA levels were not elevated in our transcriptional profile (Khan et al., 2017). We examined spatial expression of modSP by HCR, which was upregulated in experimentally induced winner cells in an undamaged disc but was not upregulated in Zone A of a regenerating disc (Fig. S7I-K). To assess activation of Toll signaling, we

quantified the amount of Dorsal immunostaining present in the nuclei of cells inside and outside Zone A (Fig. S7L). Given that the antibodies against Dorsal and Wg were both generated in mice, Zone A was marked by a newly generated transgenic line, *2xHA-Myc*, in which a 2xHA-tag was added to the N terminus of the endogenous *Myc* gene (Fig. S7L'). While some Dorsal translocated into the nucleus inside of the Zone A cells (Fig. S7M-N,O), its location remained cytoplasmic in the cells outside of Zone A (Fig. S7M'-N',O). To ask whether SPE is required for death in the 8-cell zone, we quantified TUNEL in regenerating discs heterozygous for the *SPE^{SK6}* mutation (Fig. S7P,R,S) or expressing *UAS-SPE-RNAi* (Fig. S7Q,T,U). We did not find reduction of cell death surrounding the high-Myc area in either condition.

Taken together, these data suggest that increased *Myc* in the regeneration blastema leads to something closer to cell competition than super-competition, given the absence of cell death when *Myc* levels were equalized, the increased proteotoxic stress, and the presence of autophagy. Additionally, *Myc*-induced cell death outside the high-Myc zone is not identical to canonical cell competition, which is mediated by *Xrp1* and can involve *Azot* expression or Toll signaling activation. Therefore, we conclude that the cell death seen around the *Wg+*, high-Myc Zone A does not fall neatly into the types of cell competition characterized through experimentally induced clones.”

We tried several methods of blocking the *Myc*-induced cell death, aside from reducing *Myc*, which had other effects on regeneration.

(Ji et al., 2021 doi: 10.7554/eLife.61172) showed that flies homozygous for H99 deletion that removes *reaper*, *hid*, and *grim* did not show cell competition. However, given that H99 is on the third chromosome and that the genetic ablation system is also on the third chromosome, we were unable to obtain homozygous H99 ablated wing discs. We assessed discs that were heterozygous for H99, but cell death was not reduced in the loser cell zone. (Data upon request)

We tried to block apoptosis in the putative loser cells by overexpressing p35, using *lexAop-p35* driven by *GMR26E03-lexA*. This *GMR26E03-lexA* line has been reported to be expressed in the hinge cells (Worley et al., 2022). However, in our hands the expression overlapped with the wing pouch cells (see images above). Thus, P35 overexpression driven by *GMR26E03-lexA* also occurred in the cells targeted for ablation, creating a large swath of “undead cells” and an abnormal blastema. We have not yet identified a *LexA* driver that is restricted to the hinge. (Data upon request).

Thus, while blocking the cell death outside of the high *Myc* zone to assess the effects on regeneration is a very interesting question, the experiment is not currently feasible.

Note that we do not claim that the cell competition is beneficial for regeneration, and our findings do not depend on its being beneficial. It is possible that this cell competition is yet another example of pro-regeneration signaling or factors having a negative side effect, as we have shown for JNK signaling, which disrupts posterior cell fate (Schuster and Smith-Bolton 2015), and high *Myc* expression, which disrupts margin cell fate (Abidi et al., 2023). Eliminating this cell competition would determine the extent to which it is positive, negative, or neutral. However, as noted above, we have not found an experimentally feasible way to do this.

Minor points:

The section title for figure 6 (line 386) is not what the authors tested, since their experiment tested the outcome of the reduction in *Myc* levels with the hypomorph. A more accurate title should be along the lines of the reduction of *Myc* increases cell death in *Wg*-expressing cells and decreases in adjacent *Wg*-negative cells.

We have changed the title to “**Differences in *Myc* expression induce cell death at the periphery of Growth Zone A**” (line 389)

Reviewer 3: This study seeks to define the growth regulatory roles of *Myc* and *Tor* in a fly wing disc regeneration model that utilizes a *Gal4/Gal80* temperature- controlled reaper transgene.

The translation (OP-puromycin), EdU, and pH3 experiments in the first half of the figures provide a series of readouts for the effect of Myc and Tor alleles in the second part of the paper. The experiments confirm that reducing the genetic dose of Myc or Tor individually impair local proliferation and growth. Comparing readouts of each pathway indicate that Myc and Tor pathways exhibit reciprocal dependence on the other, although the mechanism(s) of this crosstalk is not investigated.

The authors also address the interesting question of whether blastema cells with elevated Myc use cell competition to eliminate adjacent cells at the border of the pouch. Establishing a physiologic role for cell competition during regeneration would be a significant advance. The data indicate that Myc expressing blastema cells are able to eliminate nearby "Myc-low" cells, but do not progress to testing whether this competitive apoptosis is required for efficient wing regrowth or patterning.

We agree that the question of whether this death is beneficial, neutral, or deleterious for regeneration is interesting. As noted above, we were unable to find a way to eliminate this death independently of eliminating Myc expression in order to ask this question.

As it stands, the paper relies on genetics to infer relationships between Myc-Tor and between competition and regeneration. Tests of each of these linkages would enhance mechanistic insight.

1) Myc promotes expression of ribosomal subunit proteins and RNAs. Tor promotes ribosome assembly by activating S6/S6k. Do alleles of these genes synergistically impair regrowth and, at a protein level, do Myc and Tor collaborate to produce and assemble more ribosomes in blastema cells?

We tried to build a line that has both *Myc^{PO}* and *Tor^{2L1}*. Unfortunately, the two alleles seem to be synthetically lethal - animals of this genotype failed to hatch. This is not surprising because both Myc and Tor signaling are important for normal development.

We also tried feeding rapamycin to *Myc^{PO}* animals and found inconsistent reduction of Tor activity, which we presume is due to reduced food intake by the *Myc^{PO}* larvae.

Since we were unable to ask this question experimentally, we have adjusted the conclusion to remove the word "additively".

Lines 29-32 "Here, we find that regenerative growth in imaginal discs is controlled by the transcription factor Myc and by Tor signaling, which drive proliferation and translation in the regeneration blastema."

Lines 78-79 "In this paper, we demonstrate that the transcription factor Myc and the Target of Rapamycin (Tor) signaling pathway drive blastema growth."

2) Is cell competition required for efficient or accurate regeneration? The data show that competition occurs, but it could be a simple clearance mechanism to remove damaged cells rather than a selection process to ensure regrowth by the fittest cells. Blocking competition should reveal what role it plays in wing regeneration.

See comments in the response to reviewer 2. The only way we found to block the Myc-induced cell death was reduction of Myc itself, which has profound effects on regeneration. Furthermore, we do not claim that the cell competition is beneficial, and consider it equally likely to be deleterious.

Minor point:

1. The authors need to be careful in drawing comparisons between gene expression in the Reaper regeneration systems used in this study and the Eiger regeneration system used in the Worley and Floc'hlay single cell seq studies. Reaper is a cleaner death signal that acts immediately above caspases, while Eiger triggers Hid/Reaper but also activates JNK, which primes regeneration by activating AP1 transcriptional

activity. Regeneration after Eiger-death exhibits some differences with Reaper, including a stronger ecdysone response. Its ok to make comparisons between the Reaper and Eiger systems, but there are caveats that should be mentioned.

This is a key point that we now discuss. Lines 543-549.

“**Interestingly**, these distinct zones were not identified by scRNA-seq experiments (Floc’hlay et al., 2023; Worley et al., 2022), **suggesting that** the cells in Zones B and C may not have differential gene expression. **However, these studies induced damage through overexpression of Eiger, whereas we induced damage through overexpression of Reaper, and the blastema after Eiger-induced ablation may not have different zones of Myc expression and Tor activity.**”

Second decision letter

MS ID#: dev.204760R1

MS TITLE: Myc and Tor drive growth and cell competition in the regeneration blastema of *Drosophila* wing imaginal discs

AUTHORS: Felicity Ting-Yu Hsu and Rachel K. Smith-Bolton

Dear Rachel,

I have now received all the referees reports on the above manuscript, and have reached a decision. The referees' comments are appended below, or you can access them online: please go to .

The overall evaluation is positive and we would like to publish this manuscript in Development. Reviewers #1 and 2 recommend acceptance, whereas Reviewer #3 requests additional experiments. I ask that you address points by Reviewers #2 and 3 as you revise a final version of the manuscript. Revisions can be addition of new experiments that address the points, or discussion and clarification in the text of the manuscript that addresses the points. To be clear, including additional experiments is not required. Please detail how you have addressed the changes in a point-by-point response to me. If you do not agree with any of their criticisms or suggestions explain clearly why this is so. Let me know if you have questions.

Reviewer 1

Advance summary and potential significance to field

This study uses the *Drosophila* wing imaginal disc to elucidate genetic and cellular mechanisms of tissue regeneration, in particular how signals within and outside the blastema drive growth. The authors find that both Myc and Tor crosstalk to regulate cell proliferation, protein translation and ribosome biogenesis using a variety of cell biological, genetic, and pharmacological assays. In doing so, they discover distinct zones within the blastema that they characterize in terms of their rates of cell proliferation, death, Myc and Tor expression, as well as protein translation. Interestingly, the zones appear to evoke a similar paradigm to cell competition, where in the cell expressing Myc outcompetes and induce enhanced apoptosis in neighboring cells. Overall, this is a robust and rigorous study revealing a deeper understanding of how growth signals are coordinated to regulate regeneration, which are likely conserved in other models.

Comments for the author

The authors revision addressed all concerns by this reviewer.

Reviewer 2*Advance summary and potential significance to field*

This manuscript describes the spatial relationship between the expression patterns of factors that are important for regeneration Wg and Myc with the distribution of phosphorylated S6 and OPP-detected protein translation, which they use as evidence that the blastema is partitioned into zones. The authors subsequently relate myc expression to their cell death observations as cell-cell competition between the zone expressing Wg/Myc and the surrounding cells in the adjacent zone that lack Wg/Myc expression.

The messages of this manuscript are the characterization of wing disc regeneration into zones based on the expression patterns of wg, myc, cell proliferation phosphorylated S6 and OPP-detected protein translation, and the possible importance of cell-cell competition in regeneration of these zones. In my opinion, the novelty in the manuscript can come from the potential for describing how cell-cell competition and cell fitness can be involved in structuring the blastema and in making sure everything is regenerated with the correct organization. The authors provide new evidence that cell-competition mechanisms are involved with the data in supplemental figure S7.

Points for revision:

I believe that needs a little more evidence that cell-cell competition is important for publication in Development is the provision of sufficient evidence that Myc-regulated cell-cell competition is important for the regeneration of the zones and possibly the subsequent wing structure. As I indicated in my first review, the concepts of the blastema having zones of gene expression, of cell proliferation, of Tor in size control, etc. have been published. I feel the novelty comes from the possibility that Myc-regulated cell competition is involved in regenerating specific zones that the authors identified in this manuscript. The authors observed that Myc-hypomorphs increase apoptosis in wingless-expressing zone, so they inferred that Myc-mediated cell competition occurs. However, there are known connections between growth and apoptosis, and Myc is important for growth, so the observations in the manuscript may not involve an active competition mechanism. The authors now provide some evidence for cell competition: the decrease in apoptosis in a Flower (Fwe) mutant in Figure S7 is promising support. However, the authors state these results may be due to a background regeneration response. The authors could resolve this issue by testing whether their Flower mutant impairs the apoptosis response in regenerating zones that is caused by the Myc-P0 hypomorph, which they observed in Figure 6I and 6J. Also, the authors can use the Fwe and Azot mutants they have in hand to test whether impairing cell competition during regeneration leads to abnormal patterning or size differences before and after metamorphosis. I feel these experiments can provide critical evidence for the authors' conclusions that Myc-based cell-cell competition is important for the zone organization during regeneration and that regeneration of these zones is important for the contribution of the wing disc. The authors already have generated results from these mutants for Figure 6 and Figure S7R,S, and the described fly experiments could be done relatively quickly. Lastly, the evidence that cell competition occurs and restores patterning of wing disc is important for the cell competition conclusion, so this evidence should be presented in the main figures and not as supplement data.

Reviewer 3*Advance summary and potential significance to field*

The paper supports the finding that Myc-Tor collaborate to upregulate translation within the regenerative blastema of genetically injured wing discs. Additional evidence suggests that zones of Myc and Tor activity are not completely overlapping, which generates concentric rings within the regenerate that display different levels of proliferation/translation. The paper is somewhat descriptive overall, but still delivers a significant advance vis a vis translation in the blastema. I suspect that a subsequent study will use ribosome profiling to identify which mRNA transcripts are targeted for more translation by this Tor/Myc mechanism.

The authors put forth a good faith experimental effort to address my two main requests, which were to try to provide mechanistic insight into (1) how Myc and Tor collaborate to elevate translation (I suggested they look at ribosome #s) and (2) whether their hypothesized Myc-dependent cell competition is actually significant enough to affect the overall efficiency of regeneration. At least one other reviewer had the same issue with the Myc competition claims in the first submission, and in response, the authors have wisely toned down their language on this element of their conclusions. Their replies to my comments on the Myc-Tor synergy indicate that they tried some experiments, but were stymied by synthetic lethality of myc/tor trans-heterozygotes. This is an interesting result, and makes me even more curious as to the underlying mechanism. One suggestion would be to use GFP-tagged ribosome subunits to define levels/distribution of translational machinery across the disc +/- injury.

Please add a "fibrillar" label to all the nucleolus panels in Fig2,3 and S3.

Second revision

Author response to reviewers' comments

These additional suggestions from the reviewers were extremely helpful in pointing out where we needed to clarify our logic and writing. We have made the necessary changes where noted below, and pointed out which suggested experiments would not answer the questions posed and which are beyond the scope of this work.

Reviewer 1: SUMMARY OF THE ADVANCE MADE IN THIS PAPER AND ITS POTENTIAL SIGNIFICANCE TO THE FIELD:

This study uses the *Drosophila* wing imaginal disc to elucidate genetic and cellular mechanisms of tissue regeneration, in particular how signals within and outside the blastema drive growth. The authors find that both Myc and Tor crosstalk to regulate cell proliferation, protein translation and ribosome biogenesis using a variety of cell biological, genetic, and pharmacological assays. In doing so, they discover distinct zones within the blastema that they characterize in terms of their rates of cell proliferation, death, Myc and Tor expression, as well as protein translation. Interestingly, the zones appear to evoke a similar paradigm to cell competition, where in the cell expressing Myc outcompetes and induce enhanced apoptosis in neighboring cells. Overall, this is a robust and rigorous study revealing a deeper understanding of how growth signals are coordinated to regulate regeneration, which are likely conserved in other models.

SUGGESTIONS TO AUTHORS:

The authors revision addressed all concerns by this reviewer.

Reviewer 2: SUMMARY OF THE ADVANCE MADE IN THIS PAPER AND ITS POTENTIAL SIGNIFICANCE TO THE FIELD

This manuscript describes the spatial relationship between the expression patterns of factors that are important for regeneration Wg and Myc with the distribution of phosphorylated S6 and OPP-detected protein translation, which they use as evidence that the blastema is partitioned into zones. The authors subsequently relate myc expression to their cell death observations as cell-cell competition between the zone expressing Wg/Myc and the surrounding cells in the adjacent zone that lack Wg/Myc expression.

The messages of this manuscript are the characterization of wing disc regeneration into zones based on the expression patterns of wg, myc, cell proliferation phosphorylated S6 and OPP-detected protein translation, and the possible importance of cell-cell competition in regeneration of these zones. In my opinion, the novelty in the manuscript can come from the potential for describing how cell-cell competition and cell fitness can be involved in structuring

the blastema and in making sure everything is regenerated with the correct organization. The authors provide new evidence that cell-competition mechanisms are involved with the data in supplemental figure S7.

Points for revision:

I believe that needs a little more evidence that cell-cell competition is important for publication in Development is the provision of sufficient evidence that Myc- regulated cell-cell competition is important for the regeneration of the zones and possibly the subsequent wing structure. As I indicated in my first review, the concepts of the blastema having zones of gene expression, of cell proliferation, of Tor in size control, etc. have been published. I feel the novelty comes from the possibility that Myc-regulated cell competition is involved in regenerating specific zones that the authors identified in this manuscript. The authors observed that Myc-hypomorphs increase apoptosis in wingless-expressing zone, so they inferred that Myc-mediated cell competition occurs. However, there are known connections between growth and apoptosis, and Myc is important for growth, so the observations in the manuscript may not involve an active competition mechanism.

The Myc hypomorph (*Myc^{P0}*), which eliminates Myc expression in the wing imaginal disc, leads to an increase in apoptosis in the *Wg⁺* zone as the reviewer noted, but more importantly leads to a decrease in apoptosis outside the *Wg⁺* expressing zone, which is the evidence that supports the existence of cell competition. Thus, this experiment does support a general need for Myc for cell survival as the reviewer hypothesizes, as that would mean an increase in apoptosis everywhere.

We have emphasized the uniform loss of Myc in lines 418-419

“To determine whether this apoptosis was due to elevated Myc expression, we quantified TUNEL-positive cells in *Myc^{P0}* regenerating discs that lacked Myc expression throughout the disc (Fig. 6F,F',G, S2B).”

We have clarified our interpretation of the experimental results in line 423.

“This reduction in cell death outside of the *Wg⁺* zone when Myc is removed suggests that the cell death is due to Myc expression level differences.”

We have added the point that loss of Myc may make the cells of the inner zone loser cells. Lines 426-429.

The increased cell death within the *Wg⁺* zone in *Myc^{P0}* regenerating discs suggests that Myc is important for survival of these blastema cells. Given that JAK/STAT signaling can also induce cell competition (Rodrigues et al., 2012) and that JAK/STAT signaling is elevated outside the *Wg⁺* zone (Fig. 4J), the loss of Myc may render the cells inside the *Wg⁺* zone loser cells, causing the observed increase in apoptosis.

The authors now provide some evidence for cell competition: the decrease in apoptosis in a Flower (*Fwe*) mutant in Figure S7 is promising support. However, the authors state these results may be due to a background regeneration response.

We do propose that the decrease in overall apoptosis in the *Fwe* mutant may be due to an overall need for *Fwe* for cell survival. We have added the possibility that it is impacting Myc-induced competition outside Zone A and Jak-STAT-induced competition inside Zone A, as the reviewer hypothesizes. Lines 491-494.

“While this global reduction in cell death may indicate an overall role in cell survival, it may also indicate a role in Myc-induced competition outside Zone A and JAK/STAT- induced cell competition inside Zone A.”

The authors could resolve this issue by testing whether their Flower mutant impairs the apoptosis response in regenerating zones that is caused by the Myc- *P0* hypomorph, which they observed in Figure 6I and 6J.

We do not believe this experiment would distinguish between the two possibilities - an overall role in survival causing the apoptosis or a specific role in possible JAK/STAT- induced cell competition and loser cell death.

Also, the authors can use the Fwe and Azot mutants they have in hand to test whether impairing cell competition during regeneration leads to abnormal patterning or size differences before and after metamorphosis. I feel these experiments can provide critical evidence for the authors' conclusions that Myc- based cell-cell competition is important for the zone organization during regeneration and that regeneration of these zones is important for the contribution of the wing disc. The authors already have generated results from these mutants for Figure 6 and Figure S7R,S, and the described fly experiments could be done relatively quickly.

The Azot mutants did not reduce Myc-induced cell death, and so would not enable us to ask whether loss of this cell death impacts regeneration.

The Fwe mutant only reduced cell death slightly, not enough to ask whether loss of this cell death impacts regeneration.

Lastly, the evidence that cell competition occurs and restores patterning of wing disc is important for the cell competition conclusion, so this evidence should be presented in the main figures and not as supplement data.

As suggested by the reviewer, we have moved that data showing a positive result (the Fwe heterozygotes) to Figure 7 (7L, M) while leaving the complete data in the supplemental figure.

Reviewer 3: The paper supports the finding that Myc-Tor collaborate to upregulate translation within the regenerative blastema of genetically injured wing discs. Additional evidence suggests that zones of Myc and Tor activity are not completely overlapping, which generates concentric rings within the regenerate that display different levels of proliferation/translation. The paper is somewhat descriptive overall, but still delivers a significant advance vis a vis translation in the blastema. I suspect that a subsequent study will use ribosome profiling to identify which mRNA transcripts are targeted for more translation by this Tor/Myc mechanism.

We have added this possibility to the discussion. Lines 573-575

This future exploration may include determining which genes are controlled by the Myc transcription factor and which mRNAs are experiencing enhanced translation due to the increase in ribosomal activity.

The authors put forth a good faith experimental effort to address my two main requests, which were to try to provide mechanistic insight into (1) how Myc and Tor collaborate to elevate translation (I suggested they look at ribosome #s)

We did not realize that the reviewer was asking us to quantify ribosome numbers with this original comment: “Do alleles of these genes synergistically impair regrowth and, at a protein level, do Myc and Tor collaborate to produce and assemble more ribosomes in blastema cells?” This experiment is beyond the scope of this manuscript, but could be included in a future manuscript describing ribosomal profiling and changes in the mRNAs being transcribed during regeneration.

and (2) whether their hypothesized Myc-dependent cell competition is actually significant enough to affect the overall efficiency of regeneration. At least one other reviewer had the same issue with the Myc competition claims in the first submission, and in response, the authors have wisely toned down their language on this element of their conclusions.

Their replies to my comments on the Myc-Tor synergy indicate that they tried some experiments, but were stymied by synthetic lethality of myc/tor trans- heterozygotes. This is an interesting result, and makes me even more curious as to the underlying mechanism. One

suggestion would be to use GFP-tagged ribosome subunits to define levels/distribution of translational machinery across the disc +/- injury.

The Fibrillarin experiments examined ribosome production, and tagged ribosomes would not add to the analysis presented here. However, such an analysis could be included in a future manuscript describing ribosomal profiling and changes in the mRNAs being transcribed during regeneration.

Please add a "fibrillarin" label to all the nucleolus panels in Fig2,3 and S3.

We have added the label the these figure panels.

Third decision letter

MS ID#: dev.204760R2

MS TITLE: Myc and Tor drive growth and cell competition in the regeneration blastema of *Drosophila* wing imaginal discs

AUTHORS: Felicity Ting-Yu Hsu and Rachel K. Smith-Bolton

I am happy to tell you that your manuscript has been accepted for publication in *Development*, pending our standard publication integrity checks.